



# Decadal variability in extratropical Rossby wave packet amplitude, phase, and phase speed

Georgios Fragkoulidis[1]

[1]Institute for Atmospheric Physics, Johann Gutenberg University, Johann-Joachim-Becher-Weg 21, 55128 Mainz, Germany

**Correspondence:** Georgios Fragkoulidis (gfragkou@uni-mainz.de)

**Abstract.** The ongoing global, yet spatially inhomogeneous warming prompts the inspection of decadal variability in the extratropical upper-tropospheric circulation properties. This study provides observational evidence in this regard by utilizing reanalysis data to unveil past trends in the probability distribution of Rossby wave packet (RWP) amplitude ($E$), phase ($\Phi$), and phase speed ($c_p$) that may creep behind interannual variability. First, a comparison between the NE Pacific and N Atlantic

regions indicates that the 300 hPa $E$ probability distribution exhibits a seasonally- and regionally-varying decadal variability. No apparent discrepancy arises between different reanalysis datasets, except from the JJA season where the historical ones systematically underestimate $E$. Further exploiting the local in space and time character of the employed diagnostics in ERA5 reveals that, while many areas experience pronounced RWP property variations at interannual and/or decadal time scales, patterns of robust trends in the 1979–2019 period do emerge. Notably, the Northern Hemisphere $E$ field exhibits positive trends

in N Pacific, NE Atlantic, and S Asia in DJF, whereas negative trends are found in a substantial portion of the extratropics in JJA. In terms of $c_p$, distinct patterns characterize MAM, with positive trends in parts of N Atlantic and most of Europe and negative trends to the north of these regions and parts of N Pacific. The Southern Hemisphere features a poleward shift in the band of climatologically-maximum $E$ values in DJF, widespread positive $E$ trends in MAM, and positive $c_p$ trends in large parts of the extratropics in DJF and MAM. Assessing the decadal variability of RWP phase reveals zonally-extended patterns

of alternating trends in the trough-ridge occurrence ratio for MAM in the Northern Hemisphere and JJA in both hemispheres. Furthermore, no covariance is observed between area-averaged daily-mean $E$ and $c_p$ at decadal time scales, as revealed by the $E$–$c_p$ bivariate distribution trends for the different regions and seasons. Finally, it is shown that many parts of N Pacific and N America experience a shift to increasing occurrence of large-amplitude and/or quasi-stationary RWPs in DJF during 1999–2019, thus reflecting the temporal variation in trends that characterizes areas and seasons that feature pronounced variability at

interannual-to-decadal time scales.

## 1 Introduction

The extratropical upper-tropospheric circulation exhibits a substantial and impactful variability across a wide range of spatiotemporal scales. Variations in the properties of the synoptic-scale jet and wave features constitute prominent manifestations of this variability. Facilitated by the increasing data availability in recent decades, several studies have explored the causes

and effects of such variations at daily (Teubler and Riemer, 2021), weekly (Madonna et al., 2017), seasonal (Hoskins and



Hodges, 2019), interannual (Souders et al., 2014), and decadal (Simpson et al., 2014) time scales. Monitoring the interannual and decadal evolution of the atmospheric flow is particularly topical due to the ongoing global warming. The spatially inhomogeneous temperature change in past and future decades is expected to affect and be affected by the anyway naturally-varying circulation in an uncertain manner, thus reducing the confidence in regional climate projections (Shepherd, 2014). To that end,

a crucial aspect to investigate is the past decadal variability of extratropical large-scale (Rossby) wave properties, given their recurrent presence in the upper-tropospheric flow and their documented role on weather extremes (Wirth and Eichhorn, 2014; O'Brien and Reeder, 2017; Fragkoulidis et al., 2018; Röthlisberger et al., 2019; Grazzini et al., 2021; Ali et al., 2021) and teleconnections (Simmons et al., 1983; Feldstein and Dayan, 2008; Harnik et al., 2016; Branstator and Teng, 2017; Wolf et al., 2018).

Motivated by the above, previous empirical studies have addressed questions on whether and to what extent trends have emerged in the amplitude and phase speed of Rossby waves as a response to global warming. To that end, a variety of measures has been employed on reanalysis data, since waves in the upper-tropospheric flow manifest in various fields and forms. This fact — together with incompatibilities on the time series and areas under consideration — has led to discrepancies between the conclusions of these studies, such that no consensus so far emerges. A selection of key outcomes is hereafter presented.

Di Capua and Coumou (2016) provided evidence that the meandering of the 500 hPa geopotential height field over Eurasia exhibits negative trends in summer (July–September) and pronounced positive trends in autumn (October–December) between 1979 and 2015. They also found positive trends in the amplitude of quasi-stationary waves over N America year-round and more clearly in summer. Similarly, Vavrus et al. (2017) focused on the N American region and showed that the meandering of the 500 hPa geopotential height field exhibits generally positive trends in both summer and winter between 1980 and 2014.

Based on the 500 hPa geopotential field, Blackport and Screen (2020) utilized a metric rooted in the Huang and Nakamura (2016) local wave activity to examine the autumn and winter "waviness" of the Northern Hemisphere circulation from 1979 to 2018 and reported no trend in this regard. Souders et al. (2014) used the 300 hPa streamline-following envelope of meridional wind and found no trend between 1979 and 2010 for the Northern Hemisphere as a whole or sectors of it, but a positive trend for the annual Southern Hemisphere Rossby wave amplitude. On a similar note, Karami (2019) used the 250 hPa envelope of

meridional wind and reported no trend in any sector or season of the Northern Hemisphere between 1980 and 2013. Coumou et al. (2015) applied spectral analysis on the Northern Hemisphere summer 500 hPa meridional wind field and found robust reductions in the amplitude of several synoptic-scale wavenumbers, but no clear trend in the phase speed between 1979 and 2013 in the midlatitudes (35°N–70°N). Finally, Riboldi et al. (2020) assessed the Northern Hemisphere phase speed in summer and winter based on a spectral analysis of the 250 hPa meridional wind field and found no trend in the midlatitudes (35°N–

75°N) between 1979 and 2018, although robust negative trends emerged for shorter periods within.

The evolution of Rossby waves in the real atmosphere is influenced by a multitude of concomitant processes and phenomena, such that they typically materialize as eastward-propagating Rossby wave packets (RWPs) of highly-dynamic and spatially-varying properties rather than sinusoidal features (Wirth et al., 2018). In recognition of that, the present study aims to illuminate aspects of decadal variability in the extratropical upper-tropospheric circulation and provide further observational evidence in

this regard by utilizing local — in both space and time — diagnostics of RWP properties. Following Fragkoulidis and Wirth





(2020), the analytic signal of meridional wind is employed for the diagnosis of RWP amplitude, phase, and phase speed. The advantage of this approach is that it exposes the spatiotemporal evolution of RWPs, thus allowing — among other things — the analysis of local in space probability distributions of their properties. The RWP amplitude provides an estimate of the magnitude of jet meandering at synoptic scales, as this is reflected in the zonal succession of northerlies and southerlies in the upper-tropospheric wind field. The RWP phase can be used for the identification of troughs and ridges along the jet, while the RWP phase speed reflects the rate of change of their position in the zonal direction. These diagnostics are employed on reanalysis data of the upper-tropospheric wind field in order to assess the decadal variability of RWP amplitude, phase, and phase speed and report on possible trends in these regards. Going beyond the seasonal means, the long-term variability of the entire seasonal distribution of these properties is also considered, with a focus on the occurrence of large-amplitude and/or quasi-stationary wave packets.

The remainder of this article is organized as follows. The employed data and methods are presented in Sect. 2. Section 3 contains the analyses toward the aforementioned objectives. Specifically, Sect. 3.1 explores the decadal variability of the RWP amplitude probability distribution in specific regions of the Northern Hemisphere midlatitudes. Section 3.2 presents Northern and Southern Hemisphere maps of decadal trends in RWP amplitude, phase, and phase speed. Sections 3.3 and 3.4 assess trends in the bivariate probability density function of RWP amplitude and phase speed and trends in the occurrence of large-amplitude and/or quasi-stationary RWPs in the Northern Hemisphere. Finally, Sect. 4 provides a summary of the main findings and concluding remarks regarding their implications, sensitivity, and agreement — or lack thereof — with previous studies. Additional analyses and technical information are included in the Supplement.

## 2 Data and Methods

### 2.1 Data

The study primarily uses ECMWF's ERA5 reanalysis (Hersbach et al., 2020) meridional wind field, $v$, for the period 1979–2019; retrieved at 2°×2°horizontal resolution and 6-hourly temporal resolution (daily at 0000, 0600, 1200, and 1800 UTC). Although the presented analyses refer to 300 hPa, adjacent isobaric levels are also employed in order to test the sensitivity in this respect. For the years 2000–2006, ERA5.1 is used instead of ERA5, thus accounting for a technical error in the ERA5 production that also affects the upper-tropospheric wind field (Simmons et al., 2020). In order to test the sensitivity of the results to the dataset of choice, the corresponding fields from NASA's MERRA-2 (Gelaro et al., 2017) and JMA's JRA-55 (Kobayashi et al., 2015) datasets for the periods 1980–2019 and 1979–2019, respectively, are also employed. Furthermore, ECMWF's historical reanalysis datasets ERA-20C (Poli et al., 2016) and CERA-20C (member #1; Laloyaux et al., 2018) are utilized for an assessment of RWP amplitude variability across a longer time frame, i.e., 1900–2010 and 1901–2010, respectively (Sect. 3.1).





## 2.2 Diagnosis of local Rossby wave packet amplitude and phase

The diagnosis of RWP properties is based on the 300 hPa meridional wind anomaly field, $v'$, computed as the deviation of $v$ from its corresponding climatological mean value over the period 1979–2019. To that end, the climatological annual cycle of the mean $v$ at each grid point and each available time in the day (0000, 0600, 1200, and 1800 UTC) is smoothed by a Fourier series expansion and restriction to frequencies 0–4 year$^{-1}$. The $v'$ field is then spatially filtered following Fragkoulidis and Wirth (2020) (hereinafter referred to as FW20). First, the discrete Fourier transform of $v'$ at each latitude circle is filtered by an adjustable Tukey window (Harris, 1978) with soft limits at zonal wavelengths 2000–10000 km. Subsequently, possible emerged discontinuities in the meridional direction are minimized by convolving $v'$ across longitude with a Hann window of 14°length[1].

The aforementioned steps effectively smooth the $v'$ field and direct the attention to transient synoptic-scale features of the upper-tropospheric flow. Diagnosing the local in space and time amplitude and phase of wave packets formed by these features is thus facilitated. The procedure toward this end follows the FW20 methodology which is outlined hereunder.

A sinusoidal wave of amplitude $E_0$ and angular wavenumber $k_0$ formed along a $v'$ latitude circle is given by:

$$v'(x) = E_0 \, cos \, (k_0 x) \tag{1}$$

Using Euler's formula, $v'(x)$ may be expressed as the sum of two equal-amplitude complex exponentials of opposite wavenumber (spatial frequency):

$$v'(x) = \frac{E_0}{2} \left( e^{ik_0 x} + e^{-ik_0 x} \right) \tag{2}$$

This expression reflects that — as with any real-valued function — the Fourier transform of $v'(x)$ is conjugate symmetric, so either half of the spectrum is redundant. Discarding the negative part ($k = -k_0$) and doubling the positive part ($k = k_0$) of the spectrum leads to the information-preserving complex-valued "analytic" representation or *analytic signal* of $v'(x)$ (Gabor, 1946):

$$A_{v'}(x) = E_0 \, e^{ik_0 x} \tag{3}$$

The real part, $Re[A_{v'}(x)]$, is equal to $v'(x)$, while its imaginary part, $Im[A_{v'}(x)]$, is equal to the Hilbert transform of $v'(x)$ (Cohen, 1995). Being a complex-valued function, $A_{v'}(x)$ may be expressed in polar form as:

$$A_{v'}(x) = E(x) \, e^{i\Phi(x)} \, , \tag{4}$$

where $E(x)$ denotes the modulus of $A_{v'}(x)$:

$$E(x) = |A_{v'}(x)| \, , \tag{5}$$

and the angle $\Phi(x)$ denotes the argument (or phase) of $A_{v'}(x)$ within the interval $(-\pi, \pi]$:

$$\Phi(x) = Arg[A_{v'}(x)] = atan2 \, \{Im\,[A_{v'}(x)], \, Re\,[A_{v'}(x)]\} \tag{6}$$

---

[1]More information on the methodology and effect of the two spatial filtering steps can be found in Sect. 2.3 of Fragkoulidis (2019).





Consequently, transforming (2) to (3) with no loss of information and comparing to (4) reveals that the local amplitude ($E_0$) and phase ($k_0 x$) of this sinusoidal wave correspond to the modulus and argument of its analytic signal, respectively.

The procedure followed in this trivial example can be generalized to real-world wave signals, where meridional wind (anomaly) along latitude circles exhibits zonally-varying amplitude and wavenumber. To that end, the sequence $v'$ along a latitude circle is decomposed into a series of complex sinusoids of different amplitude and wavenumber by means of a discrete Fourier transform:

$$\hat{v}'[m] = \sum_{\ell=0}^{L-1} v'[\ell]\, e^{-2im\pi\ell/L} , \qquad (7)$$

where $\ell$ is the longitude index, $m$ is the angular wavenumber, and $L$ is the size of $v'[\ell]$. The analytic signal of $v'[\ell]$ is then computed by setting the power of its negative frequency components to zero and doubling the positive ones:

$$A_{v'}[\ell] = \frac{1}{L} \sum_{m=0}^{L-1} \hat{A}_{v'}[m]\, e^{2im\pi\ell/L} , \qquad (8)$$

with:

$$\hat{A}_{v'}[m] = \begin{cases} \hat{v}'[m], & \text{for} \quad m = 0,\ L/2 , \\ 2\hat{v}'[m], & \text{for} \quad 1 \le m \le L/2 - 1 , \\ 0, & \text{for} \quad L/2 + 1 \le m \le L - 1 . \end{cases} \qquad (9)$$

Subsequently, the local amplitude ($E[\ell]$) and phase ($\Phi[\ell]$) of $v'[\ell]$ are computed using (5) and (6), respectively. As a final step, $E[\ell]$ is restricted to wavelengths above 4,000 km using the aforementioned Tukey window filtering method.

Repeating the above for every latitude circle of the $v'$ field results in the two-dimensional RWP amplitude ($E$) and phase ($\Phi$) fields. Figure 1 shows the upper-tropospheric flow on 22 September 2018 0000 UTC with a view to illustrating the outcome of these diagnostics. The filtered 300 hPa $v'$ field at this instant is characterized by pronounced northerlies and southerlies organized into two main wave packets in the North Pacific and North Atlantic regions (Fig. 1a). The corresponding $E$ field marks these areas of enhanced amplitude, while the $\Phi$ field designates the ridge–trough succession in the flow (Figs. 1b,c).





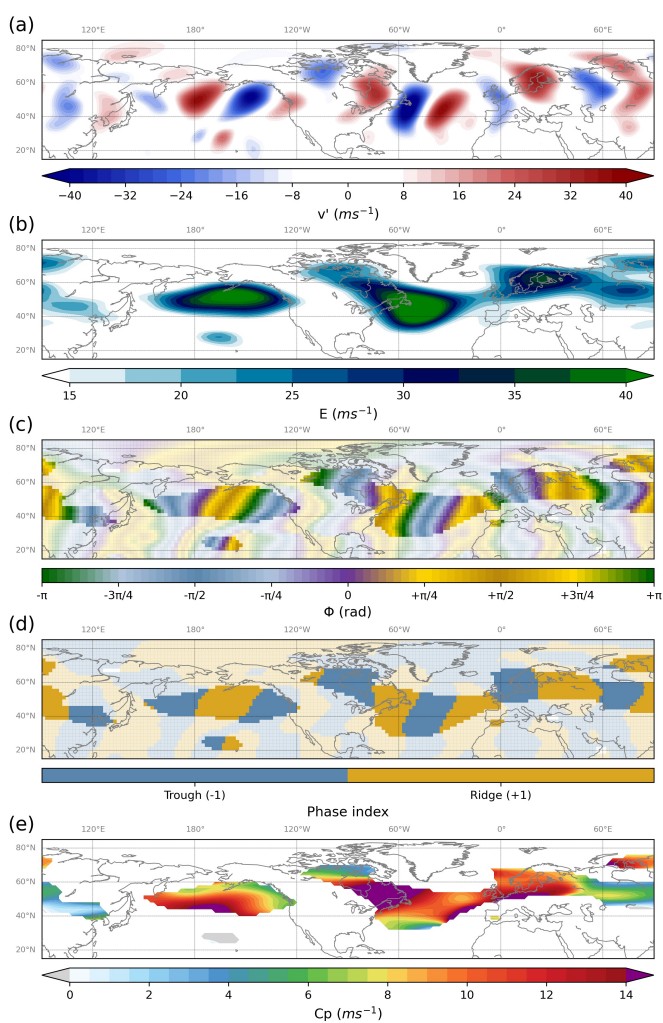

**Figure 1.** Upper-tropospheric flow diagnostics based on the ERA5 300 hPa meridional wind anomaly ($v'$) field on 22 September 2018 0000 UTC: (a) spatially-filtered $v'$, (b) amplitude ($E$), (c) phase ($\Phi$), (d) phase index (ridges: +1, troughs: -1), and (e) zonal phase speed ($c_p$). Opaque colours in panels (c) and (d) indicate the areas where RWP objects are detected.

## 2.3 Detection and amplitude of Rossby wave packet troughs and ridges

140   The information about the local amplitude and phase in the $v'$ field is further exploited toward a novel method for the detection of trough and ridge features. At first, a phase index field is constructed based on the instantaneous $\Phi$ field as follows. Phase values between 0 and $\pi$ are found in the longitudinal ranges between local maxima in southerlies and northerlies (i.e., purple and green colours in Fig. 1c, respectively), which correspond to ridges in the Northern and troughs in the Southern Hemisphere. Conversely, phase values between $-\pi$ and 0 correspond to troughs in the Northern and ridges in the Southern Hemisphere.

145   Based on this distinction, grid points within ridges in both hemispheres are assigned a phase index of +1, while grid points



within troughs in both hemispheres are assigned a phase index of $-1$. The resulting phase index field for the example of Fig. 1 is shown in panel d.

The investigation of flow features evolving within the atmospheric continuum requires the introduction of one or more thresholds to define and spatially delimit them. In order to define trough and ridge features in this study, discernible RWP structures are identified in the flow based on a local amplitude threshold of $15ms^{-1}$ as per FW20. Opaque colours in Figs. 1c,d indicate the grid points that satisfy this condition (i.e., $E \geq 15ms^{-1}$ for the given grid point and its adjacent grid points in longitude, latitude, and time) on this particular instant and thus comprise *RWP objects*. Troughs and ridges are herein defined as these areas within the RWP objects that have a phase index value of $-1$ and $+1$, respectively (Fig. 1d).

### 2.4 Diagnosis of local Rossby wave packet phase speed

When it comes to the temporal evolution of the upper-tropospheric midlatitude flow, another important property of RWPs emerges. In particular, the typically eastward motion of troughs and ridges reflects the fact that the $v'$ phase field varies in both space and time, thus giving rise to the concept of local RWP phase speed. In general, the phase function for a zonally-propagating wave of angular wavenumber $k(x,t)$ and angular frequency $\omega(x,t)$ is given by:

$$\Phi(x,t) = k(x,t)\,x - \omega(x,t)\,t + \Phi_0(x)\,, \tag{10}$$

where $\Phi_0$ is the phase at $x=0$ and $t=0$. The local in space and time zonal phase speed is thus given by:

$$c_p = \frac{\omega}{k} \tag{11}$$

$$\text{with:} \quad \omega = -\frac{\partial\Phi}{\partial t}, \tag{12}$$

$$\text{and:} \quad k = \frac{\partial\Phi}{\partial x} = \frac{1}{a\cos\phi}\frac{\partial\Phi}{\partial\lambda} \tag{13}$$

where $\lambda$ denotes longitude ($\lambda = 2\pi\ell/L$, with $0 < \lambda \leq 2\pi$), $\phi$ denotes latitude, and $a$ denotes the Earth's radius. Given the $\Phi$ field at every latitude, longitude, and time, $\omega$ and $k$ are computed at those grid points that comprise RWP objects (FW20). As is evident from Fig. 1e, the resulting $c_p$ field in this example is neither erratic nor uniform within the individual RWP objects; it exhibits gradual variation at synoptic scales. Overall, the example of Fig. 1 manifests that the upper-tropospheric wind field is generally characterized by transient synoptic-scale waves, with troughs and ridges forming RWPs of spatiotemporally-varying amplitude and phase, the investigation of which requires local in space and time diagnostics.

### 2.5 Magnitude and monotonicity of decadal trends

The linear trends in the annual time series of various metrics associated with the aforementioned RWP properties are evaluated by the Theil-Sen estimator (Sen, 1968), i.e., the median slope of lines connecting any two data points. This approach reduces the sensitivity of the trend magnitude to outliers compared to the least squares method, which is rather crucial for time series like those assessed in this study that can be characterized by pronounced interannual variability. Prior to evaluating the trend magnitude, the time series of every grid point are standardized. This allows the assessment and interpretation of the trends with



respect to the local climatology (i.e., the 1979–2019 mean of a given field and season) and interannual variability (approximated by the standard deviation of a given field's seasonal mean) .

Assuming that the analyzed time series consist of independent random variables, the statistical significance of the detected trends over time is assessed via the non-parametric Mann-Kendall test for the null hypothesis that there is no "monotonic" trend
(Gilbert, 1987; Serinaldi and Kilsby, 2016). A "monotonic" positive (negative) trend herein denotes that the variable under consideration gradually increases (decreases) with time, such that the decadal trend outweighs the year-to-year variations and the null hypothesis can be rejected at the given significance level. The seasonal means of consecutive years are expected — and indeed found — to be largely independent (i.e., the lag-1 autocorrelation is close to zero) for all examined fields in this study (not shown). Consequently, prewhitening the annual time series of every grid point leads to minor changes in the statistical
significance patterns of the resulting trends.

## 3  Results

### 3.1  Decadal variability in the Rossby wave packet amplitude probability distribution over NE Pacific and N Atlantic

To start with, the interannual and decadal variability of RWP amplitude ($E$) in specific regions of the Northern Hemisphere midlatitudes are explored. In this regard, the sensitivity to the chosen dataset is also assessed by employing an ensemble of
three *modern-era* and two *historical* reanalyses. Although all these datasets are produced using a fixed model version and data assimilation system, the modern-era ones employ observations from a dynamic array of remote and in situ measurement techniques while the historical ones are restricted to conventional surface observations (mainly pressure and marine wind). This stems from a difference in the scope of these datasets. Modern-era reanalyses primarily aim at reconstructing the best possible state of the atmosphere at any given time, while historical reanalyses primarily aim to reduce the artificial low-frequency
variability of meteorological variables induced by the ever-changing instrumentation.

Figure 3 shows the annual time series of DJF and JJA seasonal-mean $E$ at 300 hPa in the NE Pacific and N Atlantic regions (Fig. 2). Locally weighted scatterplot smoothing (LOWESS) curves (Cleveland, 1979) are constructed by time series regression (using a tricube weight function) to the $(10/N)\%$ nearest data points, where $N$ is the total amount of years available in each reanalysis dataset, i.e., 41 for ERA5 and JRA-55, 40 for MERRA-2, 111 for ERA-20C, and 110 for CERA-20C. These
smoothed curves aim to capture the decadal variability in $E$ that creeps behind interannual variability (thin solid lines in Fig. 3).

The two studied regions exhibit distinct from each other $E$ variability in DJF (Fig. 3a,b). In particular, NE Pacific is associated with a more pronounced interannual as well as decadal variability than N Atlantic, therefore multi-decadal trends may be harder to identify and more sensitive to the chosen time window. The NE Pacific year-to-year variability weakens in JJA, so the two regions become less distinct from each other in this respect. When it comes to the decadal variability, the two regions
appear to experience a steady negative $E$ trend in JJA during the past four decades, while the situation is less clear in DJF. Between 1920 and 1980, there is a positive $E$ trend in N Atlantic for both seasons and in NE Pacific for JJA. The NE Pacific in DJF is unique in this regard as well, as it features a distinctive double maximum between 1940 and 1980.

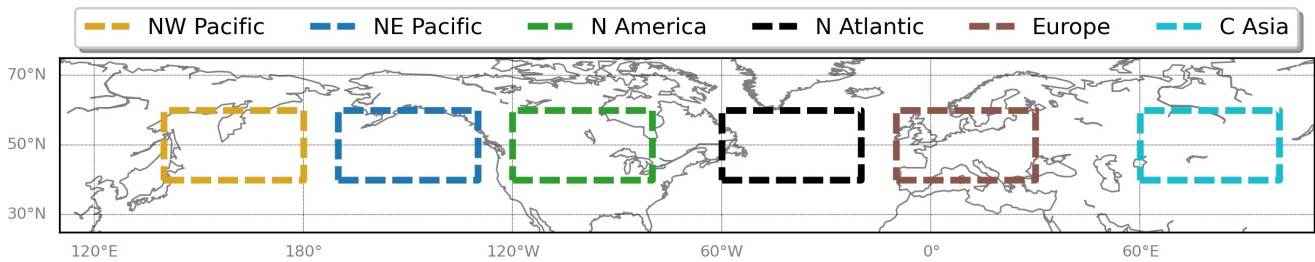

**Figure 2.** The rectangles outline six regions in the Northern Hemisphere midlatitudes where dedicated analyses are performed in parts of this study. The six regions extend from 40°N to 60°N in latitude and are restricted to the 140°E–180°, 170°W–130°W, 120°W–80°W, 60°W–20°W, 10°W–30°E, and 60°E–100°E longitudinal ranges for NW Pacific (yellow), NE Pacific (blue), N America (green), N Atlantic (black), Europe (brown), and Central Asia (cyan), respectively.

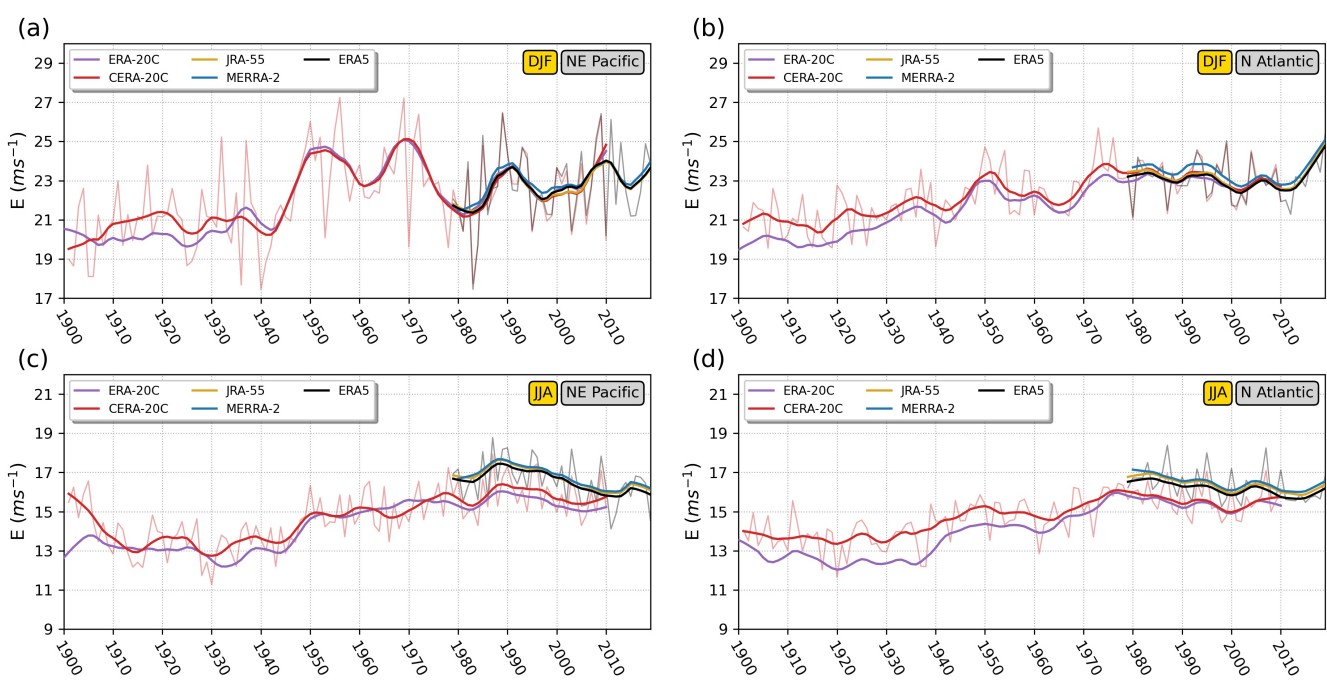

**Figure 3.** (a) LOWESS time series of DJF-mean $E$ at 300 hPa over NE Pacific based on the CERA-20C (red), ERA-20C (purple), JRA-55 (yellow), MERRA-2 (blue), and ERA5 (black) reanalysis. The thin red and black lines correspond to the original DJF-mean time series in CERA-20C and ERA5, respectively. (b) Same as (a), but for the N Atlantic region. (c,d) Same as (a,b), but for the JJA season.

The three modern-era reanalysis datasets are in close agreement with each other. The magnitude of seasonal-mean $E$ is systematically slightly higher in MERRA-2 than ERA5, while that of JRA-55 falls in between the two most of the time. During the past four decades, an agreement between modern-era and historical reanalysis datasets is evident for DJF. Interestingly, this





is not the case in JJA where the historical ones appear to systematically underestimate $E$ by about $1 ms^{-1}$ compared to the modern-era ones. This suggests that the summer upper-tropospheric circulation is not well constrained by the available surface observations in these two storm-track regions. This problem in historical reanalyses is more pronounced in the Southern Hemisphere extratropics (not shown). There are also distinct differences between the two historical reanalyses in the first half

of the 20th century. Although the more advanced CERA-20C provides a more realistic picture than ERA-20C (Laloyaux et al., 2018), exploring both showcases the changes in the $E$ annual time series that can be expected in data-sparse periods when upgrading the model version and data assimilation system. In line with the apparent correlation in seasonal-mean $E$ between all examined reanalysis datasets during the past four decades (Fig. 3), the analyses that follow are found to be insensitive to the dataset of choice. Consequently, the remainder of this study only involves the ERA5 dataset.

Apart from the central tendency (represented in Fig. 3 by the seasonal mean), it is also important to investigate the decadal variability of the entire $E$ probability distribution. Figure 4 shows the daily-mean $E$ probability density functions (PDFs) for NE Pacific and N Atlantic in DJF and JJA. Instead of individual years, the different PDFs correspond to rolling five-year periods colour-coded from blue to red corresponding to 1979–1983 and 2015–2019, respectively. This is done in order to get smoother PDFs and emphasize the decadal variability. The PDF of each five-year period is constructed based on a (non-parametric)

kernel density estimation using Gaussian kernels, the bandwidth of which is selected based on Scott's rule of thumb (Scott, 1992).

A gradual shift toward higher $E$ values is observed in NE Pacific for DJF, which is mostly evident in the lower and near-average values of the distribution (Fig. 4a). Conversely, a uniform shift toward lower $E$ values is observed for the narrower distribution of JJA (Fig. 4c). In the N Atlantic region, there are no equally apparent uniform shifts in the distribution as in

the case of NE Pacific. Specifically, no part of the DJF distribution appears to exhibit clearly visible shifts (Fig. 4b), whereas a decreasing occurrence in RWPs of above-average $E$ (i.e., RWPs with $E$ values that clearly exceed the climatological PDF maximum) is evident in JJA (Fig. 4d).

Figure 5 shows the decadal trends (based on the Theil-Sen estimator) and the corresponding 90% confidence interval for 9 percentiles of the five-year $E$ probability distributions in the two regions and seasons. It illustrates the fact that all parts

of the NE Pacific distribution exhibit positive trends in DJF and negative trends in JJA, with some variation between the percentiles that indicates changes in the distribution shape. As suggested by the visual inspection of Figs. 4b,d, the N Atlantic is characterized by lower in magnitude trends. Although the negative trends in JJA are near-uniform, DJF features an increase in the lower $E$ values and a decrease in the above-average ones such that a narrowing of the distribution arises. The generally larger confidence intervals in the percentile trends of NE Pacific compared to those of N Atlantic reflect the aforementioned

pronounced interannual and decadal variability of the former region.



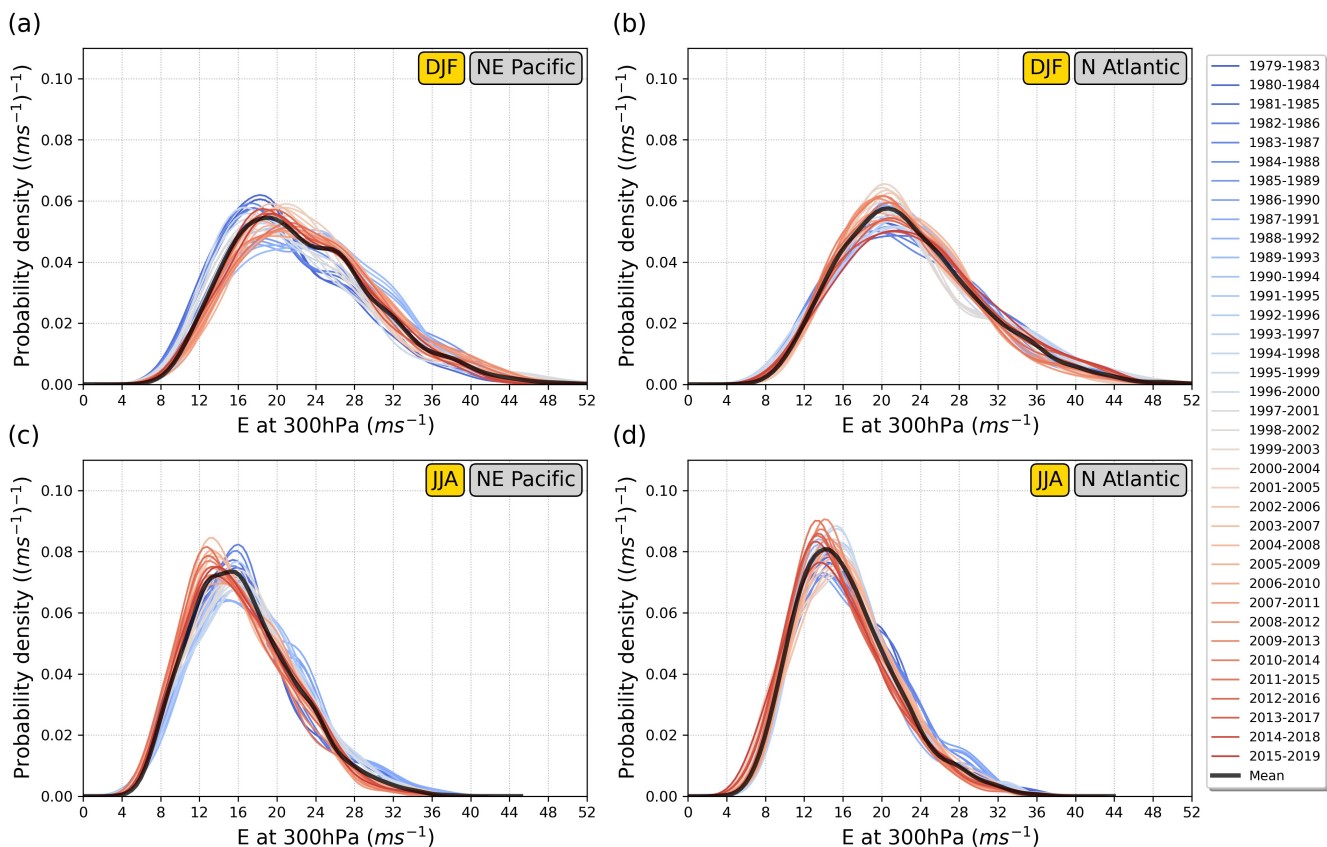

**Figure 4.** (a) Climatological (1979–2019) PDF of the DJF daily-mean $E$ at 300 hPa over NE Pacific based on ERA5 (black line). The coloured lines depict the corresponding PDFs of successive 5-year periods, as indicated in the legend to the right. (b) Same as (a), but for the N Atlantic region. (c,d) Same as (a,b), but for the JJA season.





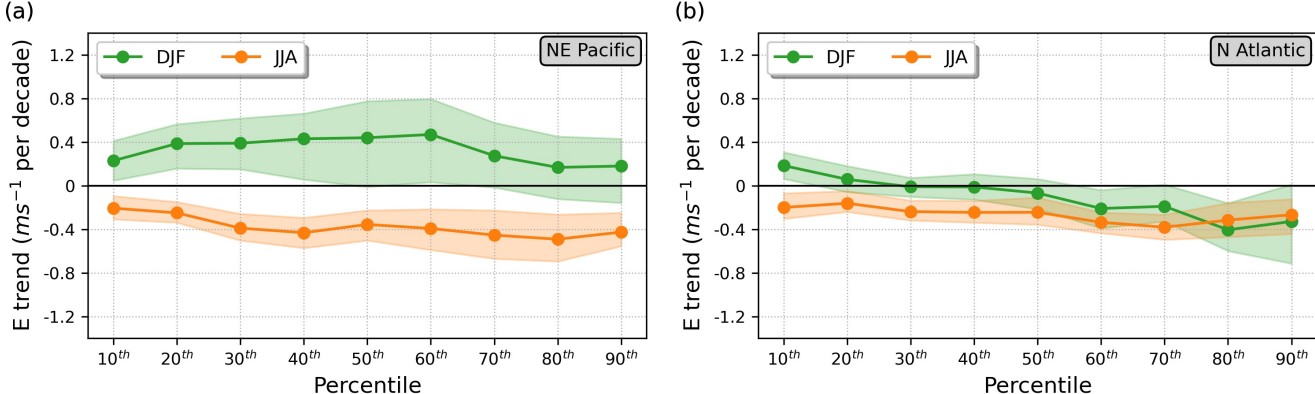

**Figure 5.** (a) 1979–2019 linear trends (Theil-Sen estimator; connected dots) in 9 percentiles of the DJF (green) and JJA (orange) NE Pacific 5-year PDFs (Fig. 4a,c) of daily-mean $E$ at 300 hPa in ERA5. The shading indicates the 90% confidence interval. (b) Same as (a), but for the N Atlantic region.

## 3.2 Decadal trends in Rossby wave packet properties

Overall, Figs. 3–5 showcase the need to explore the upper-tropospheric circulation variability using local in space diagnostics of its properties for each season separately. In this subsection, the diagnostics presented in Sect. 2 are utilized to produce maps of 1979–2019 linear trends in seasonal-mean RWP amplitude ($E$), phase ($\Phi$), and phase speed ($c_p$) at 300 hPa for the Northern

and Southern Hemisphere extratropics (Figs. 6–8). Given that the annual time series for each grid point are standardized, the decadal trends are given in units of standard deviations ($\sigma$) per decade (Sect. 2.5). The Mann-Kendall test in each grid point assesses the null hypothesis that the corresponding trend is not monotonic at the $\alpha$=0.1 significance level. Finally, overlaid in the maps are the respective multi-year seasonal means, which are more thoroughly discussed in FW20.

### 3.2.1 Amplitude

Figure 6 shows the 1979–2019 trends in standardized seasonal-mean $E$ at 300 hPa for the Northern and Southern Hemisphere extratropics, as well as the respective climatological-mean $E$. Evidently, several regions undergo monotonic trends in this regard (indicated by red hatching), while a pronounced regional and seasonal variability — as hinted in Sect. 3.1 — thereof emerges.

    The Northern Hemisphere winter season features mostly positive $E$ trends in the midlatitudes with statistically significant

values in N Pacific, NE Atlantic, and S Asia, whereas monotonic negative $E$ trends are found at higher latitudes including NE Canada, Greenland, and Siberia (Fig. 6a). A widespread negative $E$ trend is evident for the summer season with several regions in the midlatitudes exhibiting statistically significant values (Fig. 6c). Opposite trends characterize large parts of the N Pacific and N Atlantic regions in MAM, with positive and negative values, respectively (Fig. 6b). Finally, negative trends emerge in areas of N America, N Atlantic, and Eurasia in SON, but only a few parts of these regions exhibit monotonic trends (Fig. 6d).





In the Southern Hemisphere summer season (DJF), a poleward shift in the fairly zonal band of climatologically-maximum
$E$ values is apparent (Fig. 6a). Namely, there is a weakening in its subtropical edge and a strengthening in its poleward edge
and other areas of the Antarctic. Positive trends in middle and high latitudes span a large area in MAM (Fig. 6b), while less
uniform signals emerge in JJA and SON (Fig. 6c,d).

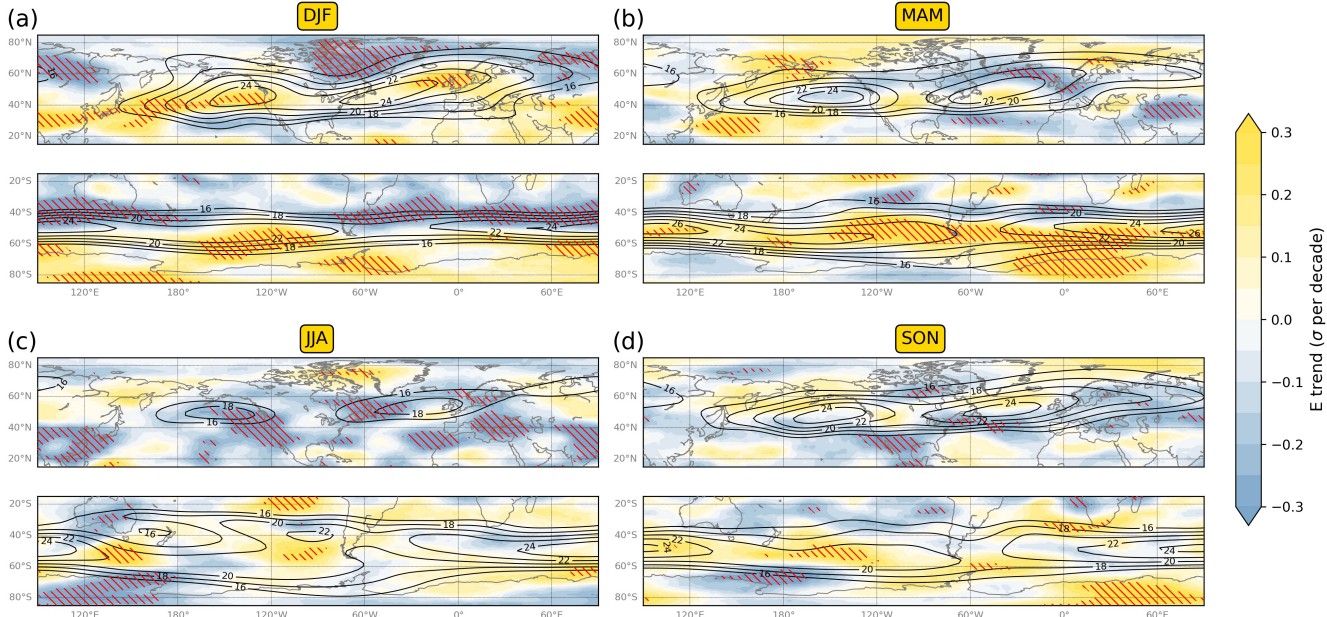

**Figure 6.** Maps of 1979–2019 linear trends (Theil-Sen estimator; colour shading) in the standardized seasonal-mean $E$ at 300 hPa for (a)
DJF, (b) MAM, (c) JJA, and (d) SON in ERA5. Red hatching indicates areas where the trend monotonicity is statistically significant at the
0.10 significance level. Black contours correspond to the climatological-mean (1979–2019) $E$ field of the respective season.

### 3.2.2   Phase

Whether a region undergoes a decadal trend in RWP amplitude or not, it is essential to examine the possibility of trends in
RWP phase as well. Even in areas where $E$ exhibits no monotonic trend, a change in the trough-ridge occurrence ratio over
time may signify changes in the local weather and climate. To that end, the seasonal-mean RWP phase index is computed based
on those time instances that feature an RWP object (Sect. 2.3). This metric can take any value between $-1$ and $+1$, indicating
whether and by how much troughs or ridges prevail in the given location and season. It is above zero when ridges occur more
often than troughs and below zero in the opposite case.

Figure 7 shows the 1979–2019 trends in the standardized seasonal-mean RWP phase index at 300 hPa, as well as the
climatological-mean phase index. The latter is based on the full phase field of $v$ rather than the RWP $v'$ field, such that
it reflects the climatological tendency toward positive or negative phase values associated with stationary waves. Since the
seasonal-mean RWP phase index at low latitudes remains undefined for seasons when no RWP object is detected, the trend is





not assessed at grid points where more than 25% of the seasons (i.e., 11 or more out of the total 41 seasons) fall in this category (masked areas in Figs. 7,8,10).

The analysis reveals that monotonic trends do emerge for certain regions and seasons. Statistically significant positive trends are found in DJF over the Arctic Ocean region that corresponds to the left exit of the N Atlantic jet stream and RWP track, indicating that the frequency of ridges is increasing relative to that of troughs (Fig. 7a). A larger part of the midlatitudes

experiences statistically significant trends in MAM (Fig. 7b). Specifically, a positive trend is found over most of Europe with negative trends upstream (N Atlantic) and downstream (Western Asia). A dipole trend pattern characterizes the Arctic ocean; probably associated with the increasing occurrence of a cross-polar wind pattern, rather than two separate processes. Finally, negative trends are found over the Sea of Japan with an equally large region of positive trends downstream over the N Pacific.

Prominent patterns emerge in JJA with statistically significant trends of alternating signs extending from N America to

Central Asia (Fig. 7c). Negative trends are found in Eastern N America, NE Atlantic, and Central Asia, while positive trends are found in the regions in between as well as in Greenland. When it comes to SON, positive trends are found in Eastern Europe and negative in Central Asia (Fig. 7d). Further to the north, negative and positive trends are found to the west and east of Greenland, respectively.

Statistically significant RWP phase trends in the Southern Hemisphere are mostly found in the central and southern sides

of the RWP tracks. The most prominent trend pattern emerges in JJA with a succession of positive and negative values in the 40°S–80°S latitude band (Fig. 7c). This "wave train" formation in the RWP phase trend pattern, which is also observed in the Northern Hemisphere, is not coincidental. If a process or interaction of processes leads to a gradual (i.e., acting on decadal time scales) increase in the frequency of troughs or ridges over a particular area, transient RWPs develop upstream and, primarily, downstream accordingly, such that a zonally-extended phase index trend pattern of alternating signs emerges.

Given that the aforementioned trends apply to the phase index of RWP objects, their interpretation should also consider trends in the RWP frequency, which generally follows the $E$ trends (not shown). A positive RWP phase index trend may not correspond to an increase in the occurrence of ridges, if it coincides with a negative trend in RWP frequency. One scenario in this case is that the reduction in troughs is larger than the reduction in ridges, such that the ratio of ridges over troughs increases.


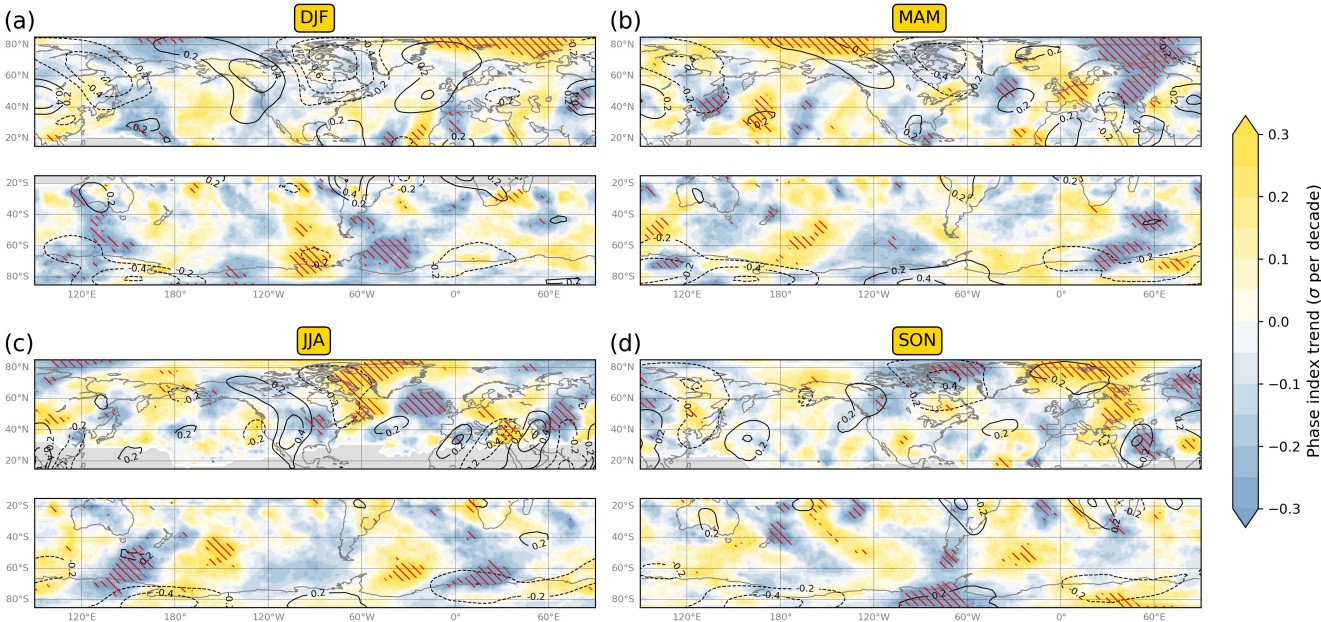

**Figure 7.** Same as Fig. 6, but for the phase index of RWP objects at 300 hPa. The black contours correspond to the multi-year mean phase index based on the filtered $v$ at 300 hPa. Masked grid points are indicated by grey shading.

### 3.2.3 Phase speed

The possibility of decadal trends in the zonal propagation speed of RWPs (i.e., $c_p$) is next explored. As in the case of the RWP phase index, the seasonal-mean $c_p$ is computed based on those time instances that feature an RWP object. Although no uniform slow-down or acceleration of upper-tropospheric troughs and ridges emerges in the Northern Hemisphere, certain regions in certain seasons do experience statistically significant trends in this regard (Fig. 8).

The most notable pattern of monotonic trends in the Northern Hemisphere occurs in MAM (Fig. 8b). The $c_p$ trend in N Pacific is generally negative, with statistically significant values over the Philippine and East China Seas; regions that also experience monotonic positive $E$ trends (Fig. 6b). In contrast, the rest of the midlatitudes feature positive $c_p$ trends, with statistically significant values over most of Europe and parts of N Atlantic and Western Russia. Finally, a band of monotonic negative trends characterizes Greenland and the Arctic Ocean region to the north of Europe.

No large-scale organized formation of statistically significant trends emerges in the other seasons, except for the negative values in Siberia in DJF and N Atlantic in JJA (Figs. 8a,c). In contrast to the Northern Hemisphere, the Southern Hemisphere extratropics exhibit a widespread tendency toward a $c_p$ increase. Statistically significant positive trends cover large parts of the S Atlantic Ocean in DJF and MAM, the Indian Ocean in DJF, and the S Pacific Ocean in MAM and JJA.




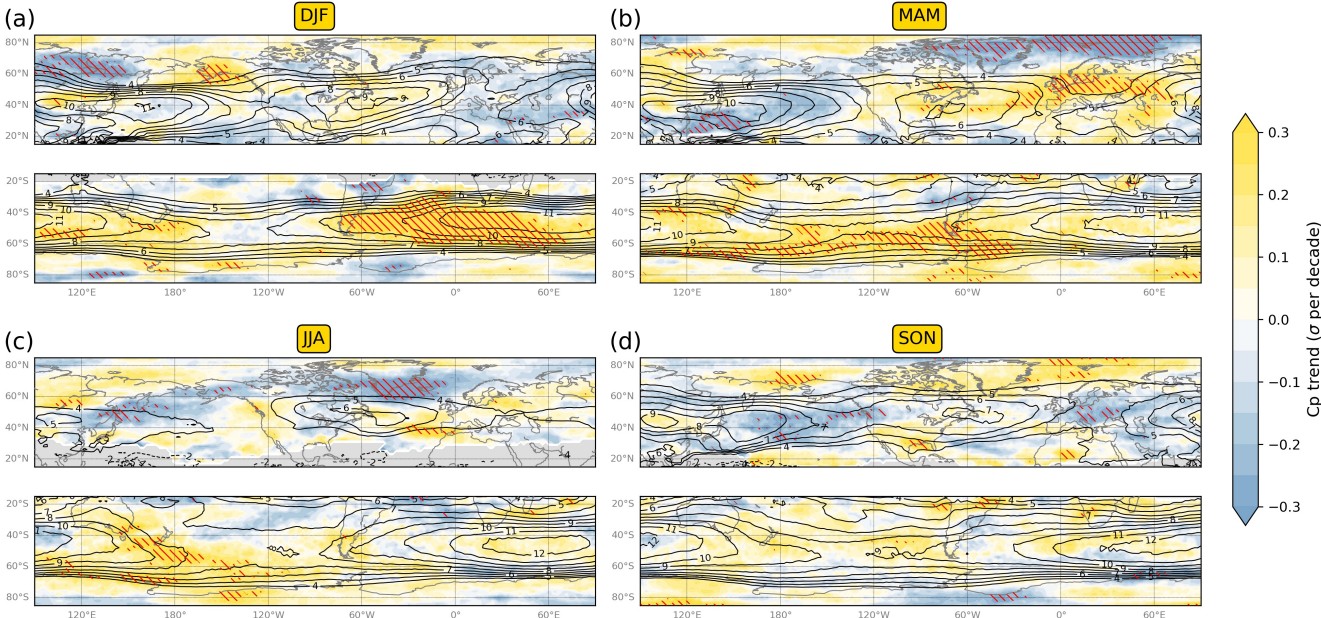

**Figure 8.** Same as Fig. 6, but for $c_p$ at 300 hPa. Masked grid points are indicated by grey shading.

### 3.3 Joint variability in Rossby wave packet amplitude and phase speed

As indicated in Sect. 3.1, seasonal probability distributions of local RWP properties may exhibit uniform shifts and/or changes in their shape. Therefore, the aforementioned seasonal-mean trend maps are not necessarily indicative of changes in e.g. the distribution tails of the associated fields. To that end, this subsection explores the 1979–2019 decadal trends in the entire $E$ and $c_p$ seasonal distributions for the six regions outlined in Fig. 2. Moreover, the distributions of these two properties are not investigated in isolation, but jointly. This allows the inspection of possible shifts in the bivariate $E$–$c_p$ distribution toward a certain regime. For example, it is worth knowing whether the negative $E$ trend over a region concerns the fast wave packets, the quasi-stationary ones, or much of the $c_p$ domain. Similarly, one can focus on a specific $E$ regime, e.g., large-amplitude RWPs, and assess the associated $c_p$ trend.

In contrast to the analysis of Fig. 4, averaging the daily-mean $E$ and $c_p$ over a region only accounts for grid points where an RWP object is detected (i.e., where $E \geq 15 ms^{-1}$; see Sect. 2.3) and, thus, $c_p$ is defined. Furthermore, days when a RWP object covers less than 10% of the region (i.e., 23 out of 231 grid points) are omitted from the $E$ and $c_p$ time series. Based on the resulting time series, Fig. 9 shows the climatological-mean and decadal trend of the daily-mean $E$ and $c_p$ bivariate PDF over the aforementioned regions in the four seasons. The underlying PDFs of the entire data sample (i.e., climatological probability distribution) and the data of each season separately are derived from two-dimensional Gaussian kernel density estimations with bandwidths based on Scott's rule of thumb. The resulting two-dimensional PDFs reflect the density of the data points in the $E$–$c_p$ domain and have units of $(ms^{-1})^{-2}$ (probability per unit of space). The probability density annual time series for each



point in the $E$–$c_p$ domain is then standardized and its decadal trend magnitude and monotonicity are evaluated as described in Sect. 2.5.

The distribution shifts emerging in Fig. 9 reflect to some extent the individual decadal trends of seasonal-mean $E$ and $c_p$ shown in Figs. 6 and 8. In a simple scenario, the bivariate PDF shift in a region where the seasonal means of both $E$ and $c_p$ exhibit a positive trend will indicate an increase in the occurrence of high-$E$ high-$c_p$ RWPs and a decrease in the low-$E$ low-$c_p$ ones. A hypothetical increase of RWPs with above-average $E$ and average $c_p$ and RWPs with above-average $c_p$ and average $E$ could also result in positive $E$ and $c_p$ trends. In light of the above, hereafter highlighted are apparent trend patterns in the PDFs that cannot be inferred from the seasonal-mean trends of the individual fields.

The positive $E$ trend in DJF over the two N Pacific regions is associated with an increase in RWPs of high $E$ and above-average $c_p$ (Figs. 9a,b). The positive $E$ trend over Europe is associated with an increase in high-$E$ average-$c_p$ RWPs and a decrease in RWPs of low $E$ and high or low $c_p$ (Fig. 9e). Although the changes are generally not statistically significant, the negative $c_p$ trend in C Asia involves RWPs of any $E$.

The negative and positive MAM $c_p$ trends in NE Pacific and N America, respectively, are mostly associated with corresponding changes in the $c_p$ distribution of near-average $E$ RWPs. In addition, the positive $c_p$ trend over Europe is associated with a decrease in high-$E$ low-$c_p$ RWPs and an increase in above-average $c_p$ occurrences for RWPs of non-extreme amplitudes (Fig. 9k). Finally, the positive $c_p$ trend in C Asia is instead primarily associated with an increase in low-$E$ high-$c_p$ RWPs (Fig. 9l).

The NW Pacific region experiences an increase in RWPs of below-average $E$ and $c_p$ RWPs in JJA, with a decreasing occurrence of RWPs of all other regimes (Fig. 9m). On the other hand, NE Pacific experiences a shift from RWPs with high $E$ and low or average $c_p$ to low-$E$ low-$c_p$ RWPs (Fig. 9n). No change is observed for N America, while the $E$ reduction over N Atlantic and Eurasia appears to favor RWPs of above-average $c_p$; a shift that is more prominent over Europe (Fig. 9o–r).

Fewer statistically significant changes in the $E$–$c_p$ distribution occur in SON compared to the other seasons (Fig. 9s–x). RWPs of low $c_p$ and above-average $E$ over NE Pacific occur more frequently at the expense of low-$E$ high-$c_p$ RWPs. Despite the generally negative trends in both seasonal-mean $E$ and $c_p$ over Eurasia (Figs. 6d and 8d), Europe exhibits a shift toward RWPs of lower $c_p$ whereas C Asia exhibits a reduction in high-$c_p$ RWPs and an increase in the average-$c_p$ ones (i.e., a narrowing of the $c_p$ distribution for most $E$ regimes).

Another key aspect emerging from the 24 investigated probability distributions is that there is no consistent underlying relation between the $E$ and $c_p$ trends, that is, there is no covariance between the two properties at decadal time scales. For example, a gradual shift toward higher RWP amplitudes is not necessarily associated with a shift toward lower or higher RWP phase speeds. Related to that, as the shape of the climatological $E$–$c_p$ spectra implies, there is weak covariance between the two properties at daily time scales as well, when all RWPs are considered (see also section 8 in the Supplement of FW20).





**Figure 9.** (a–f) Climatological (1979–2019) bivariate PDF of the DJF daily-mean $E$ and $c_p$ at 300 hPa $(10^{-3}(ms^{-1})^{-2}$; black contours) over (a) NW Pacific, (b) NE Pacific, (c) N America, (d) N Atlantic, (e) Europe, and (f) Central Asia in ERA5. Colour shading corresponds to the 1979–2019 linear trend (Theil-Sen estimator) in the seasonal $E$–$c_p$ probability density. Red hatching indicates areas where the trend monotonicity is statistically significant at the 0.10 significance level. (g–l), (m–r), (s–x) Same as (a–f), but for MAM, JJA, and SON, respectively.

## 3.4 Temporal variation of trends in mean and extreme RWP properties

As shown in Sect. 3.1, NE Pacific experiences pronounced decadal variability in the seasonal-mean $E$ field in DJF (Fig. 3). This suggests that when considering shorter periods within 1979–2019, the respective trends in RWP properties may exhibit





significant temporal variation in some regions and seasons. The objectives of this section are, first, to underline and illustrate this aspect and, secondly, to report on the associated variability in the occurrence of RWP amplitude and phase speed extremes.

Figure 10 provides an example of pronounced temporal variation in seasonal-mean $E$ and $c_p$ trends by separating the Northern Hemisphere DJF analysis into the two equally-long periods of 1979–1999 and 1999–2019. Evidently, many areas are characterized by $E$ and $c_p$ trends of different sign in these two periods. A big part of the 15–40°N latitude band — including

N Pacific, the Mediterranean, and Asia — shifts from weakly negative (during 1979–1999) to positive and often monotonic (during 1999–2019) $E$ trends. Furthermore, areas to the north of 60°N experience an apparent change from monotonic negative to weak $E$ trends. Between 40°N and 60°N, the $E$ trends remain mostly positive throughout the 41-year period. When it comes to $c_p$, a positive trend during 1979–1999 characterizes most of the Northern Hemisphere extratropics, albeit with infrequent monotonicity. In contrast, monotonic negative trends are found over parts of Asia, N Pacific and N America during the 1999–

2019 period. In particular, an unusual period of five consecutive winters between 1998 and 2002 with relatively high $c_p$ values over NE Pacific (Fig. S2) appears crucial for the noticeable trend change in this area and suggests that the scarcity of monotonic $c_p$ trends in the 1979–2019 period (Fig. 8a) is caused by variability at both interannual and longer time scales.

The last part of this study focuses on the tails of the $E$ and $c_p$ seasonal distributions, exclusively, and assesses trends in the occurrence of high-$E$ and low-$c_p$ extremes. This is motivated by the documented role of these two RWP properties on hot

and cold extremes throughout the year (Sect. 1). High-$E$ extremes are defined as days when the daily-mean RWP amplitude exceeds the climatological 90th percentile of the given grid point and year day. Similarly, low-$c_p$ extremes are defined as days when the daily-mean RWP phase speed does not exceed the climatological 10th percentile. The potentially more alarming situations when RWPs are both large-amplitude and quasi-stationary constitute a subset of the $E$–$c_p$ distribution that merits particular attention. To that end, *compound* extremes are defined as days when the daily-mean RWP amplitude exceeds the

climatological 70th percentile and the daily-mean RWP phase speed does not exceed the climatological 30th percentile. The computation of the aforementioned climatological percentiles for each day of the year is described in Appendix A.

The Northern Hemisphere DJF trends in high-$E$, low-$c_p$, and compound extremes during the 1979–1999 and 1999–2019 periods are shown in Fig. 11. The count of extremes is regarded as 0 (i.e., it is not undefined) for seasons with no detected RWP object, which is typical for low-latitude grid points. Overall, the analysis demonstrates that the aforementioned temporal

variation of trends in seasonal-mean $E$ and $c_p$ is also reflected in the justifiably spottier and less robust trend patterns of their extremes. The 1979–1999 positive high-$E$ extreme trends between 40°N and 60°N and negative trends to the north and south of this band are succeeded by generally positive trends during 1999–2019; more prominently over N Pacific, China, northwestern N America, and the Mediterranean. In the case of low-$c_p$ and compound extremes, a large part of the N Pacific and N America regions experiences weak to no trends in 1979–1999, but shifts to positive and often monotonic trends in 1999–2019. Given that

midlatitude grid points experience on average around 5 high-$E$, 5 low-$c_p$, and 4 compound extremes per winter[2], monotonic positive trends of 0.2 extremes per year may be hardly perceptible to society, albeit rather substantial in the course of 20 years. For reference, the analysis for the other seasons as well as the 1979–2019 trends for all three types of extremes are provided in the Supplement (Figs. S3–S8).

---

[2]This number varies between grid points depending on how often $E$ exceeds $15 ms^{-1}$ (Appendix A).




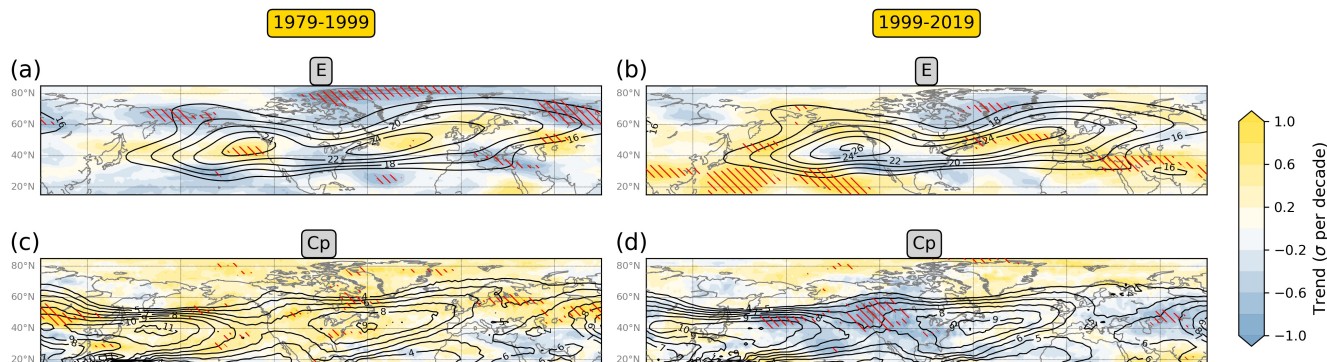

**Figure 10.** Maps of 1979–1999 linear trends (Theil-Sen estimator; colour shading) in the standardized seasonal-mean (a) $E$ and (c) $c_p$ at 300 hPa for DJF in ERA5. (b),(d) Same as (a),(c), but for 1999–2019. Red hatching indicates areas where the trend monotonicity is statistically significant at the 0.10 significance level. Black contours correspond to the multi-year mean fields of the respective period. Masked grid points are indicated by grey shading.

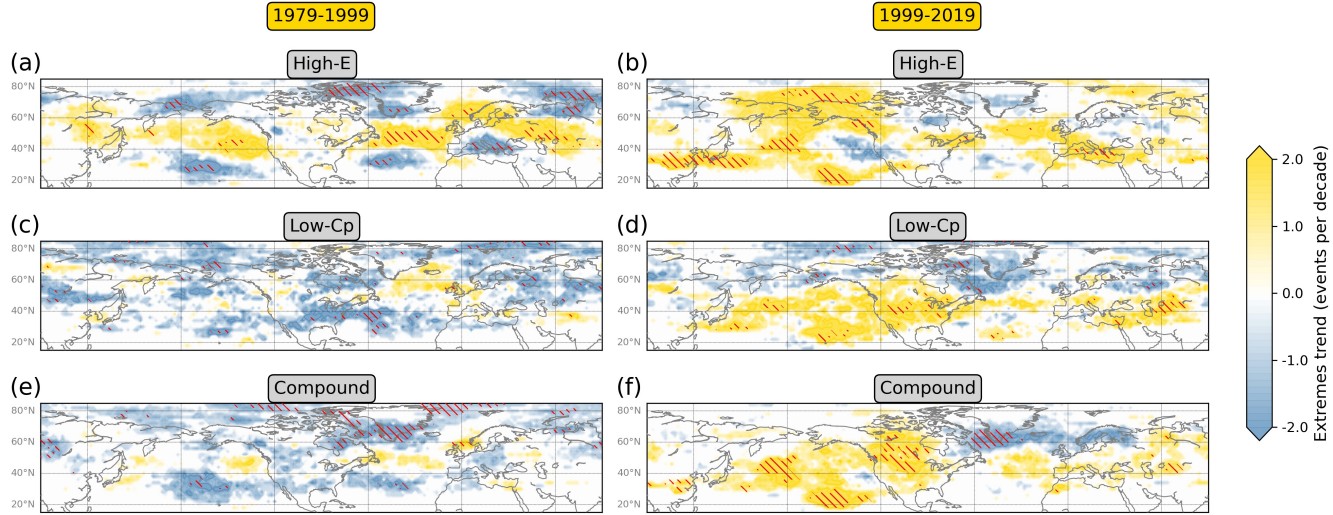

**Figure 11.** Maps of 1979–1999 linear trends (Theil-Sen estimator; colour shading) in the number (a) high-$E$ extremes, (c) low-$c_p$ extremes, and (e) compound high-$E$/low-$c_p$ extremes at 300 hPa for DJF in ERA5. (b),(d),(f) Same as (a),(c),(e), but for 1999–2019. Red hatching indicates areas where the trend monotonicity is statistically significant at the 0.10 significance level.

## 4 Summary and concluding remarks

This study assessed the decadal variability of the extratropical upper-tropospheric circulation by utilizing reanalysis data and diagnostics of local in space and time RWP amplitude ($E$), phase ($\Phi$), and phase speed ($c_p$) at 300 hPa. The main outcomes are hereafter summarized. The analysis of area-averaged seasonal-mean $E$ time series during the 1900–2019 DJF and JJA seasons




showed that NE Pacific exhibits a more pronounced interannual and decadal variability than N Atlantic, while both regions feature less interannual variability in JJA. The examined modern-era (ERA5, JRA-55, and MERRA-2) and historical (ERA-

20C and CERA-20C) reanalysis datasets were found to be in close agreement with each other regarding the seasonal-mean $E$ value, except for JJA where the historical ones systematically underestimate $E$. Focusing on the 1979–2019 period in ERA5, the decadal evolution of the daily-mean $E$ probability distribution indicated seasonally- and regionally-varying trends in its mean and shape. The decadal variability of mean RWP properties in the entire Northern and Southern Hemisphere extratropics was then explored and some notable patterns of monotonic trends are highlighted here. A substantial part of the N Pacific, NE

Atlantic, and S Asia undergoes positive $E$ trends in DJF, whereas an $E$ decrease is found in areas at higher latitudes in DJF and much of the Northern Hemisphere extratropics in JJA. The Southern Hemisphere extratropics are characterized by a poleward shift in the band of climatologically-maximum $E$ values in DJF and widespread positive $E$ trends in MAM. Zonally-extended trend patterns of alternating signs emerge in the trough-ridge occurrence ratio for MAM in the Northern Hemisphere and JJA in both hemispheres. When it comes to $c_p$, MAM features positive trends over parts of N Atlantic and most of Europe, as

well as negative trends to the north of these regions and much of N Pacific. Furthermore, high-latitude areas of the N Atlantic experience negative $c_p$ trends in JJA, while positive trends cover large parts of the Southern Hemisphere in DJF and MAM. An assessment of bivariate $E$–$c_p$ probability distribution changes demonstrated that the aforementioned Northern Hemisphere trends are associated with inconsistent — between seasons and regions — shifts in the $E$–$c_p$ domain, thus suggesting a lack of covariance between $E$ and $c_p$ at decadal time scales. As illustrated for the Northern Hemisphere $E$ and $c_p$ fields in DJF,

interannual and/or decadal variability prevail in many areas that do not exhibit monotonic trends in the 1979–2019 period. Associated with that, it was found that sizable portions of the N Pacific and N America regions experience positive trends in the occurrence of large-amplitude and/or quasi-stationary RWPs in DJF during 1999–2019.

The presented analyses do not change qualitatively when performed at the 250 hPa or 350 hPa levels instead of 300 hPa. In addition, as indicated by Fig. 3, the results remain largely unchanged when employing the JRA-55 or MERRA-2 datasets

instead of ERA5. One exception here is that JRA-55 features widespread monotonic positive $E$ trends in the Southern Hemisphere high latitudes (i.e., to the south of 60°S) in all seasons between 1979 and 2019, thus differing substantially from ERA5 (Fig. 6) and MERRA-2. The spatial variation in the $E$ and $c_p$ trends that could be unveiled through the use of local in space diagnostics (Figs. 6 and 8) implies that the analyses of sections 3.1 and 3.3 are sensitive to the size and location of the rectangles for some regions and seasons (Fig. 2). Accounting also for the pronounced temporal variation in the trends of some regions

(Sect. 3.4), it becomes evident that trends in such properties of the upper-tropospheric flow should be interpreted with caution when focusing on specific areas and/or short time periods. Finally, information regarding the advantages and limitations of the employed RWP diagnostics are provided in FW20. It is worth noting, that these diagnostics account for flow configuration of locally discernible zonal wavenumber and angular frequency. This means that they are not supposed to capture accurately the decay stage of RWPs, where wave-breaking occurs.

Due to the inherent variability in the RWP properties, differences in the employed diagnostic methods, and incompatibilities on the time periods and areas under consideration, a direct comparison with previous studies mentioned in Sect. 1 is barely possible. However, the reported widespread $E$ decrease in the Northern Hemisphere JJA season (Fig. 6c) is consistent with the



Coumou et al. (2015) spectral analysis on the midlatitude 500 hPa meridional wind field. In addition, although positive and negative $c_p$ trends do emerge in some areas of the Northern Hemisphere in DJF and JJA, there is no overall consistent trend in the midlatitudes, in agreement with the results of Riboldi et al. (2020).

The reported trends in RWP properties are most likely influenced by both externally and internally generated variability (Shepherd, 2014; Deser et al., 2016), though weighing the individual contributions was beyond the scope of this study. Moreover, the observed trend patterns in this study suggest that any future trends in the extratropical upper-tropospheric circulation — especially in the Northern Hemisphere — will not necessarily be zonally symmetric or meridionally homogeneous (see also, Simpson et al., 2014). In an effort to expose inhomegeneities in this regard, the employed methodology allowed the assessment of decadal variability and trends in RWPs without obscuring their highly-dynamic and spatially-varying properties at weather time scales. Such a weather-informed approach facilitates the investigation and interpretation of circulation-related mechanisms that influence regional climate and future projections of extreme events.

Questions regarding the causes and implications of robust trends reported in this study naturally arise. Why has the RWP amplitude been decreasing in much of the Northern Hemisphere extratropics in summer and how much has this contained, if at all, the observed positive trend in warm extremes? Do trends in the phase of Rossby waves (e.g., the negative trend over NE Atlantic in JJA) have an effect in temperature trends and the occurrence of weather extremes? What is the relation of trends in RWP properties to trends in the zonal wind component (Fig. S1)? Such investigations require and deserve dedicated studies.

*Code and data availability.* The reanalysis data used in this study have been freely available online (Sect. 2.1) and accessed as described in section 7 of the Supplement. Processed data and code employed in the presented analyses can be provided by the author upon request.

*Sample availability.* The Supplement related to this article is available online at: TBD

## Appendix A: Climatological annual cycle of $E$ and $c_p$ percentiles

The climatological annual cycles of the RWP amplitude and phase speed percentiles for a specific grid point and day of the year are computed as follows. Due to the small number of available years, each day of the year is represented by a probability distribution that comprises all daily-mean $E$ or $c_p$ values in the 21-day windows centered around it in every year. This results in a sample size of up to 861 data points (21 values from each of the 41 available years). Days when an RWP object is not identified in the given grid point are not included in the distribution, so its size decreases accordingly. The climatological $n$-th percentile is computed based on this distribution, which means that in the case of $E$ the minimum value in the distribution is $15ms^{-1}$. The climatological annual cycle of the $n$-th percentile is then constructed by repeating the above for every day of the year and it is subsequently smoothed by restriction to frequencies 0–4 $\text{year}^{-1}$ (as in the case of the climatological mean; Sect. 2.2).





*Competing interests.* The author declares that he has no conflict of interest.

*Acknowledgements.* This research has been supported by the German Research Foundation (DFG; project no. 445572993). The author acknowledges ECMWF, NASA, and JMA for providing free access to the reanalysis data and Johannes Gutenberg University Mainz for

granting computing time on the supercomputer Mogon II (hpc.uni-mainz.de). Finally, the author is grateful to Volkmar Wirth and Franziska Teubler for constructive feedback on earlier versions of the paper.



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
