# Peer review of "Decadal variability and trends in extratropical Rossby wave packet amplitude, phase, and phase speed"

_Weather and Climate Dynamics, 2022_

## Referee Comment (RC2)

**Review of manuscript "wcd-2022-28"**

Title: Decadal variability in extratropical Rossby wave packet amplitude, phase, and phase speed

Author: Georgios Fragkoulidis

**Summary and recommendation**

This study investigates the decadal variability (and trends) of midlatitude Rossby wave packet (RWP) characteristics. These RWP characteristics, being their amplitude, phase, and phase speed, are investigated for different regions and seasons using various modern-era and historic reanalysis datasets. A more detailed analysis using ERA5 assesses how the spectrum of RWP amplitude and phase speed changes during the last four decades as well as how the more extreme RWP types with large amplitude and a quasi-stationary behavior change in two subperiods of 20 years.

Overall, I found the study very interesting, thoughtful and of high quality, well-structured and well written. In my view it forms a relevant contribution to the scientific understanding in this field. That saying, I also believe the study can be improved on a few aspects. These are detailed in three general comments and several specific comments, which may be considered by the author during the revisions.

**General comments**

**1. Motivation of the study, and the interpretation and implications of the results**
The manuscript describes very clearly the applied methodology and findings of the study. However, at times I was wondering about the precise motivation as well as the main message and implications of the study. In my view, these two aspects are rather poorly addressed in the current manuscript.

For example, in the abstract, the first line briefly refers to the potential influence of global warming on decadal variability of the upper-tropospheric circulation, but this aspect is further rarely addressed in the manuscript apart from a few lines in the introduction and the last paragraph of the summary and concluding remarks. I wonder whether this study has importance for – or is related to – a range of atmospheric phenomena, including extreme weather events, climatologically wet and dry conditions, climate variability patterns of the atmosphere or coupled ocean-atmospheric system. If so, it may strengthen the manuscript to mention these aspects. Moreover, the general message of the paper and its implication(s) remain unclear to me after reading the abstract. What is the main message of this study the reader should take home?

The second point – the main message and implications - also applies to most of the results (section 3.2-3.4). The analysis is worked out well and presented clearly, however, the interpretation of the results, the context of the findings in view of the existing literature, and the implications of the new findings are missing in my view. Do the findings have, for example, any relevance to better understanding weather extremes (the large amplitude quasi-stationary waves), trends in relatively wet or dry conditions (changes/variability in troughs and ridges), interannual and decadal climate variability patterns of the climate systems, or the response of the atmospheric circulation to global warming? The two following general comments also address this aspect for sections 3.3 and 3.4, respectively.

**Analysis and presentation of the results in section 3.3**
Although the results based on Fig. 9 are potentially very interesting, the analysis section 3.3 could be further strengthened in my opinion. For example, isn't of high interest to look specifically at how slowmoving waves over North America and the Eurasian continent behave in summer in light of heat waves? How does the (lack of) positive trends in this part of the RWP amplitude – phase speed spectrum relate to findings of previous studies as for example cited in the introduction?

In addition, I found the text of this subsection quite difficult to follow. In the specific comments below are a few suggestions that may help to improve the clarity of the text.

**Analysis in section 3.4**

Although the analysis focusing on the high-E low cp RWP properties is potentially very interesting and relevant, this section left me with a few open questions and the feeling that a key element is missing. Why is the division of the analysis in two 20-year periods relevant? What do we learn from this analysis? Can we still speak about trends when the analysis is limited to 20 years or is it rather decadal variability that dominates the signal? Would even stronger and more regional signals emerge if one investigates four 10-year periods?

The focus of the second part of this subsection on high E, low cp RWP extremes is very interesting. To my surprise, however, this analysis is limited to the winter (DJF) season even though this analysis is motivated by both cold *and* hot weather extremes (as written in lines 379-380). I was expecting a discussion of these results for summer season (JJA) which is of special interest in context of heatwaves as mentioned before. Why is this not shown and discussed, apart from the note that figures for other seasons are included in the Supplement? As for the results in DJF over North America for the period 1999-2019, can these be linked to changes in cold extremes? In addition, has the author looked at trends in the extreme wave characteristics (large amplitude quasi-stationary RWPs) for the full 40-year period? Due to an increased sample size as compared to the 20-year periods, one may expect an increase of statistically significant trends in the spatial domain.

**Specific comments**

Title: The title speaks only about "Decadal variability". From the analysis throughout the manuscript, I understand that both the decadal variability *and* trends are the main foci of this paper. Therefore, I wonder whether the author has considered rephrasing the title as "Decadal variability and trends …".

Lines 1, 3, and 400. The text speaks about "decadal variability" (lines 1 and 400) and "trends" (line 3). Since both aspects are investigated jointly, would it be worth to write instead "decadal variability and trends"? See also the previous comment on the title.

Lines 4 and 200. "may creep behind" sounds a bit colloquial, please, consider rephrasing.

Line 6. Here is becomes suddenly clear that more than one reanalysis dataset is used, while the term "historical ones" remains a bit vague and unclear at this stage. Please, consider writing in lines 3-4 "… utilizing various reanalysis datasets …" and in lines 6-7 "… where 20$^{th}$ century reanalyses systematically underestimate E as compared to modern-era reanalyses." (i.e., using the same terms as in section 3.1.)

Lines 14 and 294 Is "zonally-extended patterns" really what the author means to say or perhaps "zonally-aligned" or "zonally-oriented patterns"?

Lines 19-20. "thus reflecting … … at interannual-to-decadal time scales." I cannot follow this phrase and how it is connected to the first part of the sentence. Please, consider rephrasing. In addition, wasn't the motivation of this part of the study (2$^{nd}$ half of section 3.4) linked to weather extremes (lines 379-380)? If so, then this sentence may end with addressing this aspect.

Line 28. Consider removing the word "anyways" as it sounds as colloquial word language.

Line 29-30. The first part of the sentence "To that end, … … wave properties" reads a bit difficult, please, consider rephrasing.

Line 39. Please, consider replacing "so far emerges" by, for example, "has emerged so far".

Line 39. "A selection … … presented". Please, state explicitly that you speak about previous studies.

Lines 40 and 43. The phrase "meandering of the 500 hPa geopotential height field" remains a bit vague. Do these studies address the amplitude, frequency and/or stationarity of these fields? Please, consider rephrasing in a more specific way if possible.

Line 49. Does "annual" refer here to "year-round" as in Line 42?

Line 55. Please, consider rewriting "for shorter periods within", for example, as "for shorter periods within this 40-year period".

Lines 59-60. Here, and on many occasions in the manuscript the unnecessary phrase "in this regard" or "in these regards" appears. Please, consider removing this phrase when possible.

Line 60. The phrase "local – in both space and time" reads a bit awkward. Please, consider rephrasing, perhaps as "diagnostics of spatiotemporal RWP properties". The same for the phrases "local in space and time" and "local in space" which appear on many occasions in the manuscript. I wonder whether these be rephrased, for example, as "local, transient" and simply "local", respectively.

Line 68. Please, consider rephrasing "… and report on possible trends in these regards", for example, by "derive possible trends of these characteristics."

Section 2.1. What exactly is the motivation to pick two 20$^{th}$ century reanalyses from the ECMWF? Would it be worth, instead, to use 20CR version 3 from NOAA along with CERA-20C and to leave out the more outdated ERA-20C? In addition, is there any indication to what extent the results based on CERA-20C are sensitive to the use of 1 specific ensemble member; i.e., how would results look like for different members and/or the ensemble mean? It is understood that changing the analysis accordingly may be too much to ask, especially since the focus of most of the study is on using ERA5 for the last four decades. Still, this question may occur to the reader and could proactively be addressed in the text.

Line 83. Which adjacent isobaric levels are also employed to test the sensitivity? And what came out of that sensitivity test? Later, this becomes clear in the conclusions section. I suggest replacing line 423 to this place in the manuscript.

Line 135. Please, consider rephrasing "with a view to illustrating" by "to illustrate".

Line 140. Please, consider replacing "toward" by "in".

Section 2.4. Fig. 1e illustrates the RWB phase speed at a specific time instant and shows values that range rather drastically from very low values < 4 m/s to over 14 m/s within the same troughs and ridges. Is this pointing to a weakness in the methodology or something that could be expected? Would it be worth considering spatially averaged phase speeds for the individual troughs and ridges to obtain more coherent results?

Lines 167. Please, consider replacing "manifests" by "illustrates" or "demonstrates".

Lines 171. Consider referring here explicitly to the measures introduces, and write, for example, "... of the above introduced RWB properties...".

Lines 181-182. Consider writing "... , and thus, the null hypothesis ....".

Line 188-189. Consider referring here to Fig. 2 instead of later in the text.

Line 193. Please, consider replacing "scope" by "purpose".

Line 195. What is meant by "instrumentation"? I suppose the "observational system" in general, perhaps this can be phrased as such.

Line 201. Please, the sentence "The two study regions ... ... in DJF" reads awkward, please, rephrase.

Line 204. Please, consider writing "in this season" instead of "in this respect".

Section 3.1. This section describes for the NE Pacific and N Atlantic the decadal variability and trends for DJF and JJA. Although briefly discussed in lines 206-207, a very prominent feature in Fig. 3 seems to be an increase in E throughout the 20$^{th}$ century reanalysis for both regions and seasons. Should this result be more advertised, or is there a good reason to be rather careful with such statements as these trends may be subject to artificial trends within the 20$^{th}$ century reanalysis due to the changes in the assimilated observations throughout the 20$^{th}$ century? In addition, what is the motivation to show Fig. 3 for these two specific regions only? How do the results look like for the other regions? Even if not shown, it may be briefly mentioned.

Lines 211 and 212. Please, consider replacing "ones" by "reanalyses".

Lines 212-213, "This suggests... ... storm-track regions". Please, state explicitly that this statement concerns the 20$^{th}$ century reanalyses.

Lines 214-217. Interesting, are these differences between the two 20$^{th}$ century reanalysis datasets the result of the different reanalysis products, or rather due to the uncertainty resulting from the lack of observations to constrain these reanalyses? This could be addressed using the CERA-20C ensemble, see also the earlier specific comment "section 2.1".

Line 237. The meaning of "lower in magnitude trends" is not clear to me. Does the author perhaps mean "trends of lower magnitudes"?

Line 238. The "narrowing of the distribution" seems rather a "shift in the distribution toward lower E values"; is that correct? If so, please, rephrase.

Line 239. Instead of "larger confidence intervals", would it be more accurate to say "a reduced confidence" or "larger spread in confidence" as the wider shading band indicates a reduced confidence?

Line 242. Consider to replace "using local in space diagnostics of its properties" by "using diagnostics of local RWP properties".

Lines 246-247. Please, consider writing "The Mann-Kendall test accesses at each grid point ...".

Lines 251-252. Please, remove "in this regard".

Lines 254-263. This is one of the paragraphs that make me wonder about the meaning and implications the findings, please, see general comment #1.

Lines 273-276. This sentence is unclear to me. Does it mean to say that the trends are not assessed at grid points where for more than 25% of the evaluated time intervals in the respective seasons no RWP is present? Please, rephrase for clarify.

Section 3.2.2. Specifically for variability in troughs and ridges, I wonder whether this can be linked to, for example, observed wet and dry conditions or if the patterns and their changes are for example related to circumglobal teleconnection patterns (see e.g. Fig. 7c). This results section may be strengthened by linking findings to existing literature, please, see further general comment #1.

Lines 295-299. This paragraph leaves me a bit puzzled. Are the above discussed findings still valid since changes in RWP frequencies are not included? Should changes in RWP frequencies be included in the above analysis to obtain meaningful results?

Caption of Fig. 7. Although written in the manuscript, consider saying in the caption of this figure that negative and positive values correspond to troughs and ridges for the purpose of clarity.

Lines 310-311. I suggest placing this sentence in the previous paragraph as it still discusses the Northern Hemisphere, and to start a new paragraph with the sentence "In contrast …", addressing cp changes in the Southern Hemisphere.

Lines 311-313. Interesting, these results seem to show a coherent signal. Has this also been observed in previous studies? Do we understand why this change occurs? Please, see also general comment nr. 1.

Lines 323-324. I cannot follow the sentence "In contrast to … … is defined". Please, rephrase.

Lines 336-337. I have a bit of difficulty following the rather detailed scenarios sketched here. Isn't it the general idea of the analysis to assess the changes in the RWP amplitude-phase speed spectrum, that is, how slow to fast moving waves with small to large amplitudes change? Please, consider rephrasing to improve clarity of the text.

Lines 341-342. There are several problems with the sentence "Although … … any E." For clarity, please, consider starting the sentence with "In C Asia …", refer explicitly to Fig. 9f, and consider discussing the dipole of decreased waves with high cp of any E and increased waves with low cp of any E.

Lines 339-347. These paragraphs are a bit hard to follow. Why are changes in DJF and MAM explained from the perspective of changes in E and cp, respectively, and not, for example, from both their changes? Is this because Figs 6 and 8 indicate largest changes in these seasons and regions for these specific measures respectively? If so, it may help the reader to state this. In addition, when speaking about the two N Pacific regions" (line 339) and "NE Pacific and N America" (line 343), it may be helpful for the reader to refer here to Figs 6a and 8b, respectively, to which these statements refer if I have at least understood the text correctly.

Line 348. To guide the reader, the sentence may start with "In JJA …".

Lines 16, 358, and 419; What is exactly meant with "no covariance is observed …". Does the author perhaps mean to say that no "systematic changes" are observed in trends of the RWP amplitude and phase speed spectrum?

Lines 360-361. I cannot follow the meaning of the sentence "Related to that … … in the Supplement of FW20)". Please, rephrase or consider deleting the sentence.

Line 388. I suppose the author means "not defined" or "undefined"?

Lines 405-406. Consider writing "… seasonal mean E value in DJF, but not in JJA…"

Lines 421-422. What is the implication of this finding?

Line 434. Probably, write "when" instead of "where"?

---

## Author Response (AR1)

**Decadal variability in extratropical Rossby wave packet amplitude, phase, and phase speed**
**(DOI: 10.5194/wcd-2022-28)**

**Reply to referee comments RC1 and RC2 and notes on revisions**

Georgios Fragkoulidis (JGU Mainz)

I would like to thank the two referees for the time they spent to meticulously review my manuscript. Their constructive feedback will definitely help to improve parts of the analysis and the clarity of the text. Below are point-by-point replies to their comments, where I address the raised issues and outline the respective changes to be expected in the revised manuscript. **The reviewer comments are shown in blue and the replies in black (line numbers and figure labels correspond to the initial manuscript). Where applicable, revision notes are shown in red (line numbers and figure labels correspond to the change-tracking revised manuscript).**

**Referee #1 comments (RC1)**

This research work applies a set of cutting-edge diagnostics, most of them developed by the author and previously published, to study the decadal variability and the long-term trends in Rossby wave packets and their salient properties (amplitude, phase, zonal propagation). Different analyses are performed and significant trends are discussed, with the aim to disentangle the signal of inter-annual and decadal variability from long-term trends.
The paper is clearly written and full of interesting details about the followed procedures and the results. This nice abundance of details, on the other hand, sometimes hinders the reception of the main message. I would be happy to recommend the publication of this manuscript after few additional methodological clarifications are provided and the text has been streamlined to convey key results more directly.

**General comments**

1. I have the impression that the main message of the study does not come across in a straightforward manner, "hidden" by the large amount of results. Could such a statement like "as of 2021, most trends identified in the last decades are likely the result of inter-annual-to-decadal variability. Thus, it seems that extratropical variability still outweighs almost completely any climate change signal visible in Rossby wave packets" summarize the article? Results of this paper seem to support the hypothesis that the influence of global warming on the midlatitude circulation is mostly thermodynamic and not dynamical, at least for what concerns RWPs. Something on this line is already written at lines 429–434. If this is the case, the author should state that even more clearly through the text (abstract, summary, etc. . . ).

   **Reply:** For the revised version, I will attempt to better illuminate the main outcomes of this study. Many of the referees' comments anyway point to specific reformulations needed towards this goal. Having said that, it has to be clarified that this study cannot and does not assess whether the detected trends are a forced response to global warming or the result of internal natural variability (lines 441–442) and whether any global warming influence is of thermodynamical or dynamical origin. Whether a trend in reanalysis is robust or not, not much can be concluded about the influence of global warming (e.g., the absence of a robust trend in RWP amplitude over an area may well be the result of two global warming effects that exert opposite forcings in the flow). The sentence

in lines 429–431 merely reports on the finding that some regions exhibit trends of pronounced spatiotemporal variability, so focusing on a short time period or limited area may create the wrong impression and/or lead to hasty generalizations (the relatively large amount of analyses in this manuscript is an attempt to avoid exactly that). This aspect alone does not allow inferences about the origin and extent of the global warming influence.

2. Related to previous comment, the Summary and Discussion sections could be made more to the point and easier to read by 1) splitting the body of text and discuss separately and concisely results from each analysis, using subsections and/or bullet points, or even tables or schematics; 2) enhancing the comparison between the obtained results and previous work, now limited only to two papers (lines 435-440), and citing also disagreements between results of this study and published literature.
   **Reply:** Regarding the first point, the summary of the results (lines 400–422) is indeed a bulky paragraph and not so easy to read, so in the revised version care will be taken in this regard. When it comes to comparisons with previous studies, I really struggled with this for the reasons mentioned in lines 435–437. In the end, the studies mentioned in lines 437–440 — for all their differences to this study — are the only ones I could think of where a comparison is not out of the question. Even if two studies deal with the same topic, it is not necessary that their results can be compared and agreements/disagreements can be deduced.
   **Revision:** The summary of the results is now structured as a list of outcomes from the different analyses (Lines 430–456).

3. The filtering procedure to identify RWPs is not based here on wavenumbers but on wavelengths between 2000 and 10000Km: this is good, because it reduces wavenumber aliasing due to the convergence of meridians toward the poles. However, isn't 10000 km a bit of a long wavelength for a synoptic-scale wave? What is the rationale for this choice?
   **Reply:** A wavelength of 10000 km is indeed large for typical synoptic-scale waves, but it is only chosen as the upper bound in the wavelength filtering. This choice is based on the climatological spectrum of high-pass filtered meridional wind presented in Fig. 6 of Wolf and Wirth (2017). Evidently, at 40°N most of the power is found between wavenumbers 3 and 12, which correspond to wavelengths of about 10000 km and 2500 km, respectively. Figure S1 in Fragkoulidis and Wirth (2020) (hereinafter referred to as FW20) shows the wavenumbers at each latitude that correspond to the wavelength limits (2000–10000 km), while Fig. S2 shows the effect of the "soft" filtering limits on the FFT spectrum of a 300 hPa meridional wind anomaly exemplary snapshot at 40°N . In this latitude circle, the spectral power at wavenumber 4 is fully taken into account, wavenumber 3 is weighted by a factor of about 0.65 and wavenumber 2 by a factor of about 0.05. The power at these wavenumbers is anyway typically low relative to wavenumbers 5–8, so their contribution in the filtered signal is anyway not much.

4. A RWP would lose its coherency and get deformed if its northernmost edge were to move faster than the southernmost edge, or vice versa. Given that the phase speed is expressed in m/s, and that the latitude circles have different lengths, the northern part must travel systematically slower than the southern part, and this appears to be the case looking at the meridional gradient of phase speed inside the RWP over the North Pacific in Fig. 1e. How systematic is this feature? Averaging across this gradient would result in a medium phase speed value. How can we be sure that the phase speed metric measures the actual variability in propagation and not variability in latitudinal position/extension of RWPs? This could also influence the lack of covariance between E and phase speed described in Section 3.3. Possible ways to investigate this aspect could be a systematic comparison of RWPs traveling at high and low latitudes, or a phase speed metric expressed as angular velocity.
   **Reply:** This is correct; a compactly propagating RWP (i.e., no changes in shape) has a lower phase speed in terms of $m/s$ in its northern part compared to its southern part. This is evident in the case of the N Pacific RWP in Fig. 1e, but this is not often the case. The phase speed field is typically not too homogeneous within real-world RWPs (see also reply to Specific comment #21 of Referee #2) for this and many other reasons (FW20). However, I am not sure how the phase

speed trend over an area can be affected when there is just a shift in the latitudinal position of RWPs. Does e.g. a northward shift of 5° necessarily imply a positive trend for a grid point to the north that now encounters RWPs more frequently? I don't think we can have a certain response here without knowing trends in other relevant factors as the zonal wind. Besides, the trend sign at a specific location will not be affected if its phase speed timeseries is expressed in degrees longitude per second instead of $m/s$ (this is a matter of a multiplication factor that only depends on latitude). Apologies if I haven't fully understood the concern of the referee here. I will anyway comment on this issue in the revised version, since it can indeed be confusing.

**Revision:** Figure R1 shows the phase speed field for the example of Fig. 1 (22/09/2018 00UTC) expressed in m/s (Fig. R1a) and degrees longitude per 12 hours (i.e., angular zonal phase speed; Fig. R1b). Overall, the two fields are qualitatively very similar, though some relative differences arise at higher latitudes. In an analysis that aims to track individual features (Lagrangian perspective), if a compactly-propagating RWP starts widening northward (such that it covers a larger latitudinal range), then the RWP-average phase speed may decrease and the angular phase speed remain constant. In this situation, depending on the application, one may indeed prefer to use the angular phase speed. When it comes to the phase speed trends in the Eulerian analyses presented in this study, they are identical for the two fields since the time series in every grid point are standardized. The results for the non-standardized seasonal-mean phase speed and angular phase speed fields are shown in Figs. R2 and R3, respectively. The statistical significance pattern (hatching) and the trend signs are identical between the two fields. The magnitude of the trend cannot be the same, because the fields are different. Eventually, I am hesitant to add a comment on the above in the paper, since it would neither be short nor very crucial.

[Figure]

**Fig. R1:** *(a) Map of the ERA5 zonal phase speed ($c_p$) at 300 hPa on 22 September 2018 0000 UTC (expressed in m/s). (b) Same as (a), but for the angular zonal phase speed (expressed in degrees longitude per 12 hours).*

[Figure]

**Fig. R2:** *Same as Fig. 8 in the paper, but for the non-standardized 300 hPa $c_p$ expressed in units of m/s.*

[Figure]

**Fig. R3:** *Same as Fig. R2, but for the non-standardized 300 hPa $c_p$ expressed in units of deg/12hr.*

5. Previous literature connecting Rossby waves with extreme weather often conflate together meridionally amplified waves with the occurrence of atmospheric blocking. Several studies are also based on such a tacit assumption: amplified waves, often associated with or resulting in blocking, propagate slowly and increase weather persistence. At line 433, it is said that RWP diagnostics employed in this study are not completely suited to consider the decay and wave-breaking stage of RWPs. If the diagnostics are not able to capture the effect of blocking, this might lead to misunderstandings, for instance concerning the lack of covariance between $E$ and $c_p$. How do the employed RWP diagnostics capture atmospheric blocking? Do blocking events correspond to compound high $E$/low $c_p$ events?

**Reply:** At the decay/wave-breaking stage, RWPs seize to have a clear wave structure; there is overturning and filamentary structures may appear in the flow. $E$ will typically weaken in these

cases, which is the desirable behaviour (it signifies the demise of the wave). However, the $E$ pattern may also get a bit noisy, if the meridional wind signal in the area is no longer a simple succession of northerlies and southerlies. When it comes to blocking, $E$ is typically rather high around its core, but may weaken at its outer edges if the situation resembles the aforementioned scenarios. In addition, as expected, $c_p$ will be low in blocking situations. Persistent cold extremes in Europe are associated with blocking, so the above observations are nicely showcased in Fig. 9c of FW20.

6. Related to the previous point, but on a more general level: how much are the diagnosed trends in $E$ (Fig. 6) actually due to actual increase/decrease of the amplitude of $E$, or due to a qualitative change in the structure of the waves that makes them not properly detectable by the employed methodology? (E.g., due to more/less cutoff lows or wave breaking?)
   **Reply:** This is a good question, but unfortunately I cannot think of a way to properly address it within the framework of this study and the tools at my disposal. For example, it would be interesting to investigate whether the JJA decrease in $E$ is associated with RWPs of lower amplitude throughout their lifetime and/or an increase in the frequency and/or stationarity of wave-breaking flow configurations. This possibility will be mentioned in the revised version.
   **Revision:** This issue is now more clearly discussed in Lines 467–472.

7. The lack of covariance between $E$ and $c_p$ is interesting and puzzling. The author already performed a composite analysis to understand this connection in the Supplementary Material of a previous paper (line 361). However, composites of RWP amplitude and phase speed are not obvious to interpret. Are situations with low $E$ and low $c_p$ simply associated with a weak waveguide? Is atmospheric blocking or wave breaking involved in some of those unexpected low $E$/low $c_p$ or high $E$/high $c_p$ configurations? A way to visualize more directly the underlying dynamics would be to select single days exhibiting particularly high/low $E/c_p$ and plot standard quantities during these "snapshots", as upper-level wind or geopotential.
   **Reply:** Section 8 in the FW20 Supplemental Material showed that when all days over Europe are considered a triangular shape in the $c_p$–$E$ distribution emerges. The regimes depicted in Figures S20 (a) and (c) correspond to the extreme ends of a branch in this triangle (connecting the orange and blue rectangles) where $E$ and $c_p$ are anticorrelated (i.e., lower $c_p$ is associated with higher $E$). What complicates this picture and weakens the covariance between the two properties are cases when both $E$ and $c_p$ are relatively low, the extreme ones of which are shown in Fig. S20b. This is not a strange regime. When the background flow is weak (as the referee rightly hypothesized), $c_p$ will most probably be weak as well, but $E$ doesn't have to be high; we do not have large-amplitude waves every day downsream of the jet exit region where $u$ and $c_p$ typically weaken. The latter cases can also involve wave breaking of course, where I expect both properties to weaken (see also reply to General Comment #5).

**Minor points**

1. Lines 148-153: an obvious question here would concern the sensitivity of the identified trends to the chosen 15 m/s threshold for E. Can the author comment on that?
   **Reply:** Statistical analyses on the RWP phase and phase speed have so far showed little to no sensitivity in the choice of this threshold. The $c_p$ trend patterns are more noisy when completely eliminating the threshold and all flow configurations are considered (which is not really surprising), but some qualitative agreement in the trend signs is maintained. For the revised version, I will conduct additional tests and explicitly report on whether and how much the results change when the threshold is equal to 10 or 20 m/s.
   **Revision:** Shown below are the seasonal-mean phase and phase speed trends for a RWP object identification threshold ($E_0$) of $10 ms^{-1}$, $15 ms^{-1}$ (as in the paper) and $20 ms^{-1}$. The results show minor qualitative differences between the 3 cases (Lines 458–459). In the case of $10 ms^{-1}$ the phase speed trend patterns are a bit noisier than the case of $15 ms^{-1}$, because flow features that do not correspond to a well-formed wave packet are also included in the sample. In the case of $20 ms^{-1}$, the phase speed trend patterns are again a bit noisier than for $15 ms^{-1}$, this time because fewer wave packet instances are considered (smaller samples in each grid point).

[Figure]

**Fig. R4:** *Maps of 1979–2019 linear trends (Theil-Sen estimator; colour shading) in the standardized seasonal-mean phase index of RWP objects at 300 hPa for (a) DJF, (b) MAM, (c) JJA, and (d) SON in ERA5 with $E_0 = 10ms^{-1}$. Red hatching indicates areas where the trend monotonicity is statistically significant at the 0.10 significance level. The black contours correspond to the multi-year mean phase index based on the filtered v at 300 hPa. Masked grid points are indicated by grey shading.*

[Figure]

**Fig. R5:** *Same as Fig. R4, but for $E_0 = 15ms^{-1}$.*

[Figure]

**Fig. R6:** *Same as Fig. R4, but for $E_0 = 20 ms^{-1}$.*

[Figure]

**Fig. R7:** *Maps of 1979–2019 linear trends (Theil-Sen estimator; colour shading) in the standardized seasonal-mean $c_p$ at 300 hPa for (a) DJF, (b) MAM, (c) JJA, and (d) SON in ERA5 with $E_0 = 10 ms^{-1}$. Red hatching indicates areas where the trend monotonicity is statistically significant at the 0.10 significance level. Masked grid points are indicated by grey shading.*

[Figure]

**Fig. R8:** *Same as Fig. R7, but for $E_0 = 15ms^{-1}$.*

[Figure]

**Fig. R9:** *Same as Fig. R7, but for $E_0 = 20ms^{-1}$.*

2. Lines 160-165: why not including meridional variations of phase to account for nonzonal RWP propagation, too?
   **Reply:** Although the zonal phase speed typically dominates the meridional one in the upper-tropospheric midlatitude flow, the latter would also be desirable to have in cases when the background flow is not zonally-oriented. It is however not straightforward to diagnose by using the meridional wind phase field. It would require zonal wind anomaly variations along longitudes to be taken into account and/or assumptions to be made in order to define a background flow and compute the analytic signal of its non-zonal streamlines. I have not found a good solution so far.

3. Line 177: how is the standard deviation computed? From 6-hourly or daily data, i.e., from all 3610 days, of DJF?
   **Reply:** The standard deviation here applies to the 41 seasonal means of the given field. It is meant as a simple measure of interannual variability. I also tried using the average absolute difference between successive seasonal means as a measure of interannual variability and the trend patterns were virtually the same.

4. Lines 212-213: Very interesting hypothesis, could the author elaborate more on this reduced connection between surface and upper-levels during summer with respect to the other seasons? Has this aspect been discussed by previous literature, and/or is based on some dynamical reasoning?
   **Reply:** I have not encountered any study in this regard and I could only speculate about this interesting issue. I therefore decided just to report on it as a noteworthy observation.

5. Lines 293-294: Wouldn't a simpler explanation be, that a shift in the RWP genesis consequently shifts of the same phase angle also downstream troughs and ridges? As it is phrased now, it seems like the change itself in ridge/trough occurrence is inducing a development of transient RWPs.
   **Reply:** A change in the preferred genesis region of RWPs is indeed one possibility, and a scenario that is not excluded by the more general statement in lines 293-294.

6. Lines 295-300: the interpretation of these phase plots is a bit difficult. Would it help to overlay regions of significant $E$ trends?
   **Reply:** I think overlaying $E$ trends may obscure the phase index trend map, without necessarily helping the interpretation. Perhaps my reply to Specific Comment #43 of Referee #2 clarifies the issue for Referee #1?

7. Lines 305-309: Trying to understand positive phase speed trends in MAM: could they be related to a delayed break-up of the stratospheric polar vortex?
   **Reply:** I am not familiar with observed trends in the stratospheric polar vortex. Is it a fact that in the past 4 decades it tends to maintain its strength deeper into spring and mostly over Europe? Another relevant aspect — if the polar vortex — is involved is that trends in zonal wind (Fig. S1) are a bit different than those of phase speed.

8. Line 325: the threshold on grid points penalizes features located at low latitudes, as a RWPs with a given size can satisfy the criterion if it propagates at high latitudes and not at low latitudes. It would be better to use a criterion based on the 10% of the area occupied by a RWP object inside a given region.
   **Reply:** This is correct. In the revised version the analysis will take this into account.
   **Revision:** The analysis now considers the area occupied by the RWP objects as suggested by the reviewer and the threshold is 10% of the total region area (Lines 346–347). Figures 9 and S4 (previously S2) have been updated, with hardly any change.

9. Lines 375-377: Very interesting observation, it reminds of the phase speed trends analysis of Riboldi et al. (2020). Would there be a qualitative agreement between the two metrics if the phase speed were to be averaged across all the boxes, or across a broad latitude range, to obtain a single value?
   **Reply:** In principle yes, I would expect some qualitative agreement. I am not sure about the weather time scales, though. Before comparing them I would make sure they try to represent the same component of the flow. This means, e.g., that the local phase speed metric should not only be defined when $E$ exceeds 15m/s, but at a bigger portion of the midlatitude flow.

10. Line 454: Fig. S1 is interesting and potentially connected to phase speed trends, but it is not cited or discussed anywhere in the paper. It feels a bit out of place in the closing sentence. Maybe the short Section 3.2.3 (phase speed) is a more appropriate place to introduce and briefly discuss it.
    **Reply:** It is definitely a valid point that Fig. S1 needs to be referenced a bit earlier in the text; indeed the phase speed section will be a proper fit for that.
    **Revision:** Figure S5 (previously S1) is now mentioned in Lines 332–334. Not much can be discussed about it though; just speculations. Trends in phase speed are not expected to simply follow those of zonal wind in a direct way.

**Technical/Typos**

1. Lines 7-8: sentence too long and weirdly structured, the first sentence can be removed and start with "While many areas..."
   **Reply:** The first sentence serves a couple of important purposes. It emphasizes that the diagnostics

used in this study are local in space and time and the rest of the analyses will utilize the ERA5 dataset. I will, nevertheless, try to simplify this sentence.

**Revision:** Unfortunately, I did not come up with a way to simplify this sentence; any change I could think of would change the precise meaning. I could use dashes instead of commas to "separate" the phrase "while many areas...time scales", but this would interrupt the thought more dramatically than intended.

2. Line 9: what is meant by "patterns of robust trends"?
   **Reply:** This term implies that the detected robust trends are in many cases not sparse and randomly distributed around the globe, but form spatial patterns.

3. Lines 19-20: this last sentence is very general and not clear, please reformulate.
   **Reply:** I will reformulate this. It merely meant to say that areas experiencing pronounced variability at interannual-to-decadal time scales (in one of the considered flow properties) will also exhibit temporally-varying trends if shorter periods within 1979–2019 are considered.
   **Revision:** This sentence has been reformulated and made more specific (Lines 18–21).

4. Lines 22-24: filler sentences that can be omitted or at least shortened?
   **Reply:** These are far from "filler" sentences. This paragraph and especially these first sentences aim to smoothly introduce the motivation and topic of this study; starting from the big picture and gradually specifying.

5. Line 57 (and other locations): what is meant precisely by "highly-dynamic"?
   **Reply:** The literal meaning of the word "dynamic" is implied here, i.e., changing with time.

6. Line 119: what is meant by "Arg" and "atan2"? Please use standard mathematical notation.
   **Reply:** "Arg" denotes the argument in complex analysis and it is capitalized because in this case only the principal value needed (i.e., the angle in the $(-\pi,\pi]$ interval) (line 118). The function atan2 (two-argument variant of arctangent) is used for the calculation of this argument. This will also be defined in the revised version.
   **Revision:** Footnote 2 now provides the definition of atan2.

7. Line 310: what is meant by "organized formation"? Could this sentence be shortened simply to "No significant trends"?
   **Reply:** By "organized formation" in this instance, I meant a large-scale area of consistently robust trends. I will try to come up with a more specific term.
   **Revision:** The term "organized formation" is replaced by "pattern" (Line 328).

8. Line 370: is "monotonic" here to be intended as "significant"?
   **Reply:** Yes. This point raises an issue in the text. Statistically significant trends are sometimes referred to as "statistically significant", others as "robust", and others as "monotonic". I will attempt to make these statements a bit more consistent.
   **Revision:** The term "robust" has been replaced by "statistically significant" from the discussion of the results of this study in order to make the text more consistent in this regard. Besides, the term "monotonic trend" is introduced and defined in section 2.5.

9. Lines 387-399: the take-home message of this analysis could be emphasized by a summarizing sentence at the end of the paragraph.
   **Reply:** Noted.
   **Revision:** Lines 426–428 now provide a take-home message for the resuts of subsection 3.4.

10. Line 421: "Associated with that": the connection with the previous sentence is not obvious, because the sentence above emphasized the role of inter-annual and decadal variability. This sentence, on the other hand, speaks of long-term trends.
    **Reply:** This is related to the comment about the last sentence of the abstract. In my opinion, the temporal variation of shorter trends within 1979–2019 is a manifestation of enhanced interannual-to-decadal variability.

11. Line 447: in which sense is this approach "weather-informed"? In what sense is it more related to weather than the other studies cited at lines 35-55?
    **Reply:** This is indeed not the best term and I will rephrase. I did not mean to contrast to other studies, but to suggest that climate studies can benefit from tools originally developed with a primary focus to better understand weather processes and systems.
    **Revision:** This has now been rephrased to better communicate the desired message (Lines 485–487).

12. Line 450: is "contained" to be intended as "limited", "attenuated"?
    **Reply:** It means "limited" or "held back".

**Referee #2 comments (RC2)**

**Summary and recommendation**

This study investigates the decadal variability (and trends) of midlatitude Rossby wave packet (RWP) characteristics. These RWP characteristics, being their amplitude, phase, and phase speed, are investigated for different regions and seasons using various modern-era and historic reanalysis datasets. A more detailed analysis using ERA5 assesses how the spectrum of RWP amplitude and phase speed changes during the last four decades as well as how the more extreme RWP types with large amplitude and a quasi-stationary behavior change in two subperiods of 20 years.

Overall, I found the study very interesting, thoughtful and of high quality, well-structured and well written. In my view it forms a relevant contribution to the scientific understanding in this field. That saying, I also believe the study can be improved on a few aspects. These are detailed in three general comments and several specific comments, which may be considered by the author during the revisions.

**General comments**

1. **Motivation of the study, and the interpretation and implications of the results:**
   The manuscript describes very clearly the applied methodology and findings of the study. However, at times I was wondering about the precise motivation as well as the main message and implications of the study. In my view, these two aspects are rather poorly addressed in the current manuscript. For example, in the abstract, the first line briefly refers to the potential influence of global warming on decadal variability of the upper-tropospheric circulation, but this aspect is further rarely addressed in the manuscript apart from a few lines in the introduction and the last paragraph of the summary and concluding remarks. I wonder whether this study has importance for — or is related to — a range of atmospheric phenomena, including extreme weather events, climatologically wet and dry conditions, climate variability patterns of the atmosphere or coupled ocean-atmospheric system. If so, it may strengthen the manuscript to mention these aspects. Moreover, the general message of the paper and its implication(s) remain unclear to me after reading the abstract. What is the main message of this study the reader should take home?

   The second point — the main message and implications — also applies to most of the results (section 3.2-3.4). The analysis is worked out well and presented clearly, however, the interpretation of the results, the context of the findings in view of the existing literature, and the implications of the new findings are missing in my view. Do the findings have, for example, any relevance to better understanding weather extremes (the large amplitude quasi-stationary waves), trends in relatively wet or dry conditions (changes/variability in troughs and ridges), interannual and decadal climate variability patterns of the climate systems, or the response of the atmospheric circulation to global warming? The two following general comments also address this aspect for sections 3.3 and 3.4, respectively.

   **Reply:** To start with, I agree with the referee that the abstract misses to provide a more general message. I will work on this during the revision. Apart from that, I need to clarify a few things regarding the structure and scope of this study. Its motivations are presented in the first paragraph. These constitute the overarching goals and primary reasons why I embarked on this study. The objectives of the study, however, need to be well-defined research questions involving specific action items that can deliver specific results. In this study, the objectives and action items are described in lines 58–70. Whatever follows aims to address and potentially provide answers to those specific questions based on what the data shows. The results are summarized in lines 400–422, while some concluding remarks and take-home messages based on these results are given in lines 435–453. These conclusions maybe do not sound that fascinating, but this is what came out from this study; I really have no other message for the readers. I do think that the upper-tropospheric flow plays a role in the topics the referee mentioned and together with colleagues in previous studies we have indeed provided some evidence in this regard. The analyses of this study, however, did not go in this direction and I did not deem valuable to overinterpret the results and speculate on their possible linkages with extreme weather or other climate indices. For example, I do think that the

increase in compound high-$E$/low-$c_p$ extremes over N America during the 1999–2019 winters has possibly been associated with an increase in persistent temperature extremes over the region. This linkage was clearly not assessed in this study and I wonder if writing this sentence would have any scientific value. Moreover, some readers may jump right to the conclusions of the paper, focus on this seemingly fascinating one-liner, misunderstand the fact that it is mere speculation, and perhaps start citing the paper for the wrong reasons.

**Revision:** A more general message of the study is now included at the end of the abstract (Lines 22–23).

2. **Analysis and presentation of the results in section 3.3:**

Although the results based on Fig. 9 are potentially very interesting, the analysis section 3.3 could be further strengthened in my opinion. For example, isn't of high interest to look specifically at how slow-moving waves over North America and the Eurasian continent behave in summer in light of heat waves? How does the (lack of) positive trends in this part of the RWP amplitude - phase speed spectrum relate to findings of previous studies as for example cited in the introduction?

In addition, I found the text of this subsection quite difficult to follow. In the specific comments below are a few suggestions that may help to improve the clarity of the text.

**Reply:** The referee here suggests another interesting research question regarding the linkage between slow-moving waves and heat waves over N America and Eurasia at decadal time scales. As I implied before, this would require a series of dedicated analyses that cannot really be merged with the scope of this study in what I believe is an already extensive manuscript. For the revised version I will reformulate the sentences the two referees mentioned and address some aspects of section 3.3 that needed clarification. Finally, I will re-examine whether some of the results in this section can be compared to those of previous studies.

3. **Analysis in section 3.4:**

Although the analysis focusing on the high-E low cp RWP properties is potentially very interesting and relevant, this section left me with a few open questions and the feeling that a key element is missing. Why is the division of the analysis in two 20-year periods relevant? What do we learn from this analysis? Can we still speak about trends when the analysis is limited to 20 years or is it rather decadal variability that dominates the signal? Would even stronger and more regional signals emerge if one investigates four 10-year periods?

The focus of the second part of this subsection on high E, low cp RWP extremes is very interesting. To my surprise, however, this analysis is limited to the winter (DJF) season even though this analysis is motivated by both cold and hot weather extremes (as written in lines 379-380). I was expecting a discussion of these results for summer season (JJA) which is of special interest in context of heatwaves as mentioned before. Why is this not shown and discussed, apart from the note that figures for other seasons are included in the Supplement? As for the results in DJF over North America for the period 1999-2019, can these be linked to changes in cold extremes? In addition, has the author looked at trends in the extreme wave characteristics (large amplitude quasi-stationary RWPs) for the full 40-year period? Due to an increased sample size as compared to the 20-year periods, one may expect an increase of statistically significant trends in the spatial domain.

**Reply:** The aim of Section 3.4 is mentioned in lines 365–366. It is an analysis that complements the previous ones and attempts to demonstrate the potentially pronounced temporal variation in the trends through the DJF season in the Northern Hemisphere. Very often, climate studies analysing a dataset spanning several decades compute trends in the field of interest over rolling shorter-term windows (e.g. Riboldi et al., 2020). This allows to get a better idea of how steady/gradual a trend is and/or how sensitive to the selected start and end years. In this study, it is not possible to split the 41-year period in many ways and plot maps of the 2D fields in question for each shorter time window. I only split into two periods of 21 years as an example and compute the respective trends. I think that 10-year periods are too short windows for assessing trends. 21-year periods are again on the short side, but allow enough time for some climate signals to start emerging (e.g. there is definitely a positive temperature trend observed from 2000 to 2020). For the desired linkages

to weather extremes, I refer to the previous 2 replies. When it comes to the 41-year trends of compound extremes and the split analysis for JJA extremes, these are both already included in the Supplement (Sections 5 and 7, respectively). Many complementary analyses are only provided in the Supplement because they can be a useful reference to some, but they are not too critical for the main objectives of the study.

**Specific comments**

1. Title: The title speaks only about "Decadal variability". From the analysis throughout the manuscript, I understand that both the decadal variability and trends are the main foci of this paper. Therefore, I wonder whether the author has considered rephrasing the title as "Decadal variability and trends ...".
   **Reply:** I am also wondering about this part of the title, since trends can also be thought as a special case of decadal variability. I will reconsider the referee's advice during the revision.
   **Revision:** Following the referee's advice, the title of the study is now "Decadal variability and trends in ...".

2. Lines 1, 3, and 400. The text speaks about "decadal variability" (lines 1 and 400) and "trends" (line 3). Since both aspects are investigated jointly, would it be worth to write instead "decadal variability and trends"? See also the previous comment on the title.
   **Reply:** I have to consider this again (see my previous reply).

3. Lines 4 and 200. "may creep behind" sounds a bit colloquial, please, consider rephrasing.
   **Reply:** I am also not totally happy with this verb, but it is the closest one I found to denote what I want to say, i.e., that slowly (decadal scales) and quietly (largely unnoticed) evolving trends may emerge in the background, while interannual variability dominates the scene at shorter time scales.

4. Line 6. Here is becomes suddenly clear that more than one reanalysis dataset is used, while the term "historical ones" remains a bit vague and unclear at this stage. Please, consider writing in lines 3-4 "... utilizing various reanalysis datasets ..." and in lines 6-7 "... where 20th century reanalyses systematically underestimate E as compared to modern-era reanalyses." (i.e., using the same terms as in section 3.1.)
   **Reply:** I will reformulate this based on the referee's suggestion.
   **Revision:** The sentence in Lines 6–7 has been reformulated. The sentence in Lines 3–4 remained as is. Adding the word "various" as suggested would put emphasis on the fact that I use various reanalysis datasets, whereas I want to emphasize that I use renalysis data (instead of e.g. other observational datasets or idealized models). In addition, adding the word "various" in this intro- ductory sentence would imply that all analyses in this study involve various reanalyses datasets, which is incorrect.

5. Lines 14 and 294 Is "zonally-extended patterns" really what the author means to say or perhaps "zonally-aligned" or "zonally-oriented patterns"?
   **Reply:** Yes, "zonally-extended" is what I meant to say, in order to denote that this pattern is characterized by a substantial extent in the zonal direction.

6. Lines 19-20. "thus reflecting ... ... at interannual-to-decadal time scales." I cannot follow this phrase and how it is connected to the first part of the sentence. Please, consider rephrasing. In addition, wasn't the motivation of this part of the study (2nd half of section 3.4) linked to weather extremes (lines 379-380)? If so, then this sentence may end with addressing this aspect.
   **Reply:** I will reformulate this sentence. As mentioned in my reply to General Comment #1, this study is motivated by weather extremes among other things, but it has rather specific objectives and does not deal with weather extremes specifically.
   **Revision:** This sentence has been reformulated to make this clearer (Lines 18–21)

7. Line 28. Consider removing the word "anyways" as it sounds as colloquial word language.
   **Reply:** Noted.

**Revision:** I did not manage to find a substitute here that would not affect the meaning of the sentence. Besides, I have used the word "anyway" which I believe is fine in formal writing (the word "anyways" is indeed colloquial).

8. Line 29-30. The first part of the sentence "To that end, ... ... wave properties" reads a bit difficult, please, consider rephrasing.
   **Reply:** Noted.
   **Revision:** This is now hopefully easier to read (Lines 32–34).

9. Line 39. Please, consider replacing "so far emerges" by, for example, "has emerged so far".
   **Reply:** Noted.
   **Revision:** Changed as suggested (Line 42).

10. Line 39. "A selection ... ... presented". Please, state explicitly that you speak about previous studies.
    **Reply:** Noted.
    **Revision:** This has now been specified (Lines 42–43).

11. Lines 40 and 43. The phrase "meandering of the 500 hPa geopotential height field" remains a bit vague. Do these studies address the amplitude, frequency and/or stationarity of these fields? Please, consider rephrasing in a more specific way if possible.
    **Reply:** Noted. I will clarify that "meandering" here refers to the "waviness" of the flow, which is diagnosed based on the geometry of Z500 isohypses. It is related to the amplitude of the waves, but it is different to the notion of "amplitude" I use in this study, therefore I kept the "meandering" term.
    **Revision:** I have now clarified what the authors of these studies mean by "meandering" (Line 44; the two studies in question use the same notion and almost the same index for "meandering").

12. Line 49. Does "annual" refer here to "year-round" as in Line 42?
    **Reply:** "Annual" means that the whole year is considered as one and the analysis involves the annual-mean. I will clarify this. The term "year-round" in line 42 indicated that the analysis has taken individual seasons within the year into account as well.
    **Revision:** I have further specified the result of Souders et al. (2014) (Lines 51–53).

13. Line 55. Please, consider rewriting "for shorter periods within", for example, as "for shorter periods within this 40-year period".
    **Reply:** Noted.
    **Revision:** This has been rewritten as: "for shorter windows within this 40-year period" (Line 60).

14. Lines 59-60. Here, and on many occasions in the manuscript the unnecessary phrase "in this regard" or "in these regards" appears. Please, consider removing this phrase when possible.
    **Reply:** Noted.
    **Revision:** This phrase has now been removed or replaced in all but two instances.

15. Line 60. The phrase "local – in both space and time" reads a bit awkward. Please, consider rephrasing, perhaps as "diagnostics of spatiotemporal RWP properties". The same for the phrases "local in space and time" and "local in space" which appear on many occasions in the manuscript. I wonder whether these be rephrased, for example, as "local, transient" and simply "local", respectively.
    **Reply:** The adjective "local" is often used to denote a particular part of a time series as well. I am not sure if that is what confuses the referee. Given that these diagnostics deal with variations in both the time and space dimensions, I find it necessary at times to differentiate by using phrases like "local in space" and/or "local in time". Thinking of alternatives, I doubt "local, transient" will make this clearer. I also considered "local and instantaneous" but phase speed involves centered differences within a 12-hour time span, so it is not really instantaneous. Finally, I find "spatiotemporal" too generic.

16. Line 68. Please, consider rephrasing "... and report on possible trends in these regards", for example, by "derive possible trends of these characteristics."
**Reply:** Noted.
**Revision:** Changed to "...report on possible trends in these properties" (Line 73).

17. Section 2.1. What exactly is the motivation to pick two 20th century reanalyses from the ECMWF? Would it be worth, instead, to use 20CR version 3 from NOAA along with CERA-20C and to leave out the more outdated ERA-20C? In addition, is there any indication to what extent the results based on CERA-20C are sensitive to the use of 1 specific ensemble member; i.e., how would results look like for different members and/or the ensemble mean? It is understood that changing the analysis accordingly may be too much to ask, especially since the focus of most of the study is on using ERA5 for the last four decades. Still, this question may occur to the reader and could proactively be addressed in the text.
**Reply:** This and every other question are of course welcome and valid. I thought that using two reanalyses datasets that utilise the same observations and only differ in their model version and data assimilation technique would have some value (see lines 215–217). As a matter of fact, I first thought about using the NOAA dataset, but the 300 hPa isobar was not available (only 200 hPa) so a comparison to all the other datasets would not be straightforward. I will repeat the analysis for another random member and decide accordingly on whether downloading and employing all members is necessary. Finally, I can already mention that the ensemble-mean of meridional wind in the early decades of the 20th century is not really meaningfull (tested for 20CRv3, but I guess the same would be true with the other reanalyses); different members position troughs and ridges in not so similar locations (due to the sparcity of observations) such that their mean involves a lot of cancelations between northerly and southerly components. As a result, the ensemble-mean $E$ is particularly weak and has a pronounced positive trend as observations increase and the trough-ridge successions start to align between different members.

18. Line 83. Which adjacent isobaric levels are also employed to test the sensitivity? And what came out of that sensitivity test? Later, this becomes clear in the conclusions section. I suggest replacing line 423 to this place in the manuscript.
**Reply:** As it stands, the paragraph in lines 423–434 comes right after the summary of the results and discusses their sensitivity, limitations, and comparison to additional analyses performed. I would like to keep such commentary in one place and toward the end of the manuscript, rather than spread throughout the text. Besides, I prefer section 2.1 to focus on describing what kind of data is used and keep it clear from comments on upcoming results.

19. Line 135. Please, consider rephrasing "with a view to illustrating" by "to illustrate".
**Reply:** Noted.
**Revision:** Rephrased as suggested (Line 140).

20. Line 140. Please, consider replacing "toward" by "in".
**Reply:** Noted.
**Revision:** Replaced (Line 146)

21. Section 2.4. Fig. 1e illustrates the RWB phase speed at a specific time instant and shows values that range rather drastically from very low values ¡ 4 m/s to over 14 m/s within the same troughs and ridges. Is this pointing to a weakness in the methodology or something that could be expected? Would it be worth considering spatially averaged phase speeds for the individual troughs and ridges to obtain more coherent results?
**Reply:** This variability in the phase speed field is a manifestation of the upper-tropospheric flow. Real-world RWPs are typically not too coherent/compact structures over space and time, and this is also reflected in individual troughs and ridges. The methodology has been tested in idealised models of the atmosphere where coherent RWPs can be generated and sustained for some time (Fig. 1 in FW20). Spatially-averaging phase speed over a part of the RWP can be meaningful for some applications, but I would not assume "boundaries" at the trough-ridge edges. For example,

we cannot have a trough moving at 6m/s adjacent (upstream or downstream) to a ridge moving at 7m/s.

22. Lines 167. Please, consider replacing "manifests" by "illustrates" or "demonstrates".
    **Reply:** Noted
    **Revision:** Changed to "indicates" (Line 174)

23. Lines 171. Consider referring here explicitly to the measures introduces, and write, for example, "... of the above introduced RWB properties...".
    **Reply:** Noted.
    **Revision:** Done (Line 178)

24. Lines 181-182. Consider writing "... , and thus, the null hypothesis ....".
    **Reply:** Noted.
    **Revision:** Done (Line 189)

25. Line 188-189. Consider referring here to Fig. 2 instead of later in the text.
    **Reply:** Noted.
    **Revision:** I find it better to refer to Fig. 2 once the aim of this subsection and some necessary aspects are presented to the reader. The second paragraph is where I start describing what Fig. 2 actually shows, so I prefer not to cite it earlier.

26. Line 193. Please, consider replacing "scope" by "purpose".
    **Reply:** Noted.
    **Revision:** Replaced (Line 200)

27. Line 195. What is meant by "instrumentation"? I suppose the "observational system" in general, perhaps this can be phrased as such.
    **Reply:** Yes, by "instrumentation" I mean the measurement technology of the obserational system, as opposed to other factors affecting it (e.g. data assimilation).

28. Line 201. Please, the sentence "The two study regions ... ... in DJF" reads awkward, please, rephrase.
    **Reply:** Noted. I will.
    **Revision:** This sentence has been rephrased (Line 209)

29. Line 204. Please, consider writing "in this season" instead of "in this respect".
    **Reply:** Noted.
    **Revision:** Done (Line 212)

30. Section 3.1. This section describes for the NE Pacific and N Atlantic the decadal variability and trends for DJF and JJA. Although briefly discussed in lines 206-207, a very prominent feature in Fig. 3 seems to be an increase in E throughout the 20th century reanalysis for both regions and seasons. Should this result be more advertised, or is there a good reason to be rather careful with such statements as these trends may be subject to artificial trends within the 20th century reanalysis due to the changes in the assimilated observations throughout the 20th century? In addition, what is the motivation to show Fig. 3 for these two specific regions only? How do the results look like for the other regions? Even if not shown, it may be briefly mentioned.
    **Reply:** Indeed, I decided to be careful and not say a lot about the trends in this part of the century. They are nevertheless useful to be reported on. In the revised version I will also mention potential interesting behaviours of other regions.
    **Revision:** Motivated by the questions raised by the referee, I have included in the Supplement the corresponding plots of Fig. 3 for the other four regions of interest (Figs. S1 and S2; Line 204). Since section 3.1 aims to illustrate the spatiotemporal variability in the seasonal distribution of RWP amplitude, I believe it suffices to focus on two regions and add complementary information on a few more in the Supplement. Furthermore, the positive trends between 1920 and 1980 are now emphasized a bit more and I mention that the same is true for the other four regions as well

(Lines 216–217). It is important to highlight this trend — that so far remains undocumented (to the best of my knowledge) — but it requires a dedicated study to investigate whether it is driven by anthropogenic forcing, natural variability, changes in the quality/quantity/kind of observations or a combination of these. This question is now raised in Lines 490–492.

31. Lines 211 and 212. Please, consider replacing "ones" by "reanalyses".
**Reply:** Noted.
**Revision:** Done (Lines 222–223)

32. Lines 212-213, "This suggests... ... storm-track regions". Please, state explicitly that this statement concerns the 20th century reanalyses.
**Reply:** Noted.
**Revision:** This clarification is now provided (Line 223).

33. Lines 214-217. Interesting, are these differences between the two 20th century reanalysis datasets the result of the different reanalysis products, or rather due to the uncertainty resulting from the lack of observations to constrain these reanalyses? This could be addressed using the CERA-20C ensemble, see also the earlier specific comment "section 2.1".
**Reply:** I would assume that both factors play a role, but most of the effect comes from the different model version and data assimilation. As mentioned in the reply to Specific Comment #17, I will revisit this issue by testing additional CERA-20C members.
**Revision:** I have tested an additional CERA-20C member and a comparison with the one shown in Fig. 3 is now presented in the Supplement (Fig. S3). An interesting observation regarding that is now mentioned in Lines 228–230. As expected, both the model version and the uncertainty due to lack of observations are at play, when it comes to the differences between ERA-20C and CERA-20C. It seems that after the first 2 decades of the 20th century the two members begin to align with each other (a good indication that the ensemble spread decreases). The discrepancy, though, between ERA-20C and CERA-20C remains pronounced for several decades after that (apparently associated with changes in the model version and data assimilation).

34. Line 237. The meaning of "lower in magnitude trends" is not clear to me. Does the author perhaps mean "trends of lower magnitudes"?
**Reply:** Yes, this is what I meant. I will rephrase.
**Revision:** Corrected as suggested (Line 251).

35. Line 238. The "narrowing of the distribution" seems rather a "shift in the distribution toward lower E values"; is that correct? If so, please, rephrase.
**Reply:** Figure 5b indicates that the 10th and 20th percentiles in DJF have a positive trend while the 60th, 70th, 80th, and 90th percentiles have a negative trend. This means that the tails of the distributions come closer to each other, i.e., the distribution narrows.

36. Line 239. Instead of "larger confidence intervals", would it be more accurate to say "a reduced confidence" or "larger spread in confidence" as the wider shading band indicates a reduced confidence?
**Reply:** I prefer to stick to "larger confidence intervals" in this case. "Reduced confidence" implies that we are uncertain about something (e.g., in future projections where a high spread between models or model members indicates low confidence / high uncertainty). In this case, the confidence intervals refer to the Theil-Sen estimator of the past variability.

37. Line 242. Consider to replace "using local in space diagnostics of its properties" by "using diagnostics of local RWP properties".
**Reply:** Noted.
**Revision:** Corrected as suggested (Line 257).

38. Lines 246-247. Please, consider writing "The Mann-Kendall test accesses at each grid point ...".
**Reply:** Noted.
**Revision:** Corrected as suggested (Line 261).

39. Lines 251-252. Please, remove "in this regard".
    **Reply:** Noted.
    **Revision:** This has been replaced by "in this field" (Line 267).

40. Lines 254-263. This is one of the paragraphs that make me wonder about the meaning and implications the findings, please, see general comment #1.
    **Reply:** My reply on this concern is given below General Comment #1.

41. Lines 273-276. This sentence is unclear to me. Does it mean to say that the trends are not assessed at grid points where for more than 25% of the evaluated time intervals in the respective seasons no RWP is present? Please, rephrase for clarify.
    **Reply:** I will reformulate this. It means that grid points where at least a quarter of the 41 seasons had no RWP object occurrence are masked. It doesn't refer to the 25% of the days within a season.
    **Revision:** These part has been reformulated (Lines 290–292).

42. Section 3.2.2. Specifically for variability in troughs and ridges, I wonder whether this can be linked to, for example, observed wet and dry conditions or if the patterns and their changes are for example related to circumglobal teleconnection patterns (see e.g. Fig. 7c). This results section may be strengthened by linking findings to existing literature, please, see further general comment #1.
    **Reply:** I think the trends in the frequency of troughs and ridges can indeed be associated with more or less evident changes in local weather and teleconnections. See my reply to the General Comment #1 for my thoughts regarding such linkages.

43. Lines 295-299. This paragraph leaves me a bit puzzled. Are the above discussed findings still valid since changes in RWP frequencies are not included? Should changes in RWP frequencies be included in the above analysis to obtain meaningful results?
    **Reply:** Shown below are the trends in RWP frequency. As mentioned in line 296, the patterns are qualitatively similar to those of $E$ (Fig. 6). I will consider adding this figure in the Supplement for reference. The findings of Section 3.2.2 are valid, even without considering these trends in RWP frequency. For example, a positive trend in the seasonal-mean RWP phase index means that — on days of RWP object occurrence — the frequency of ridges increases relative to that of troughs.

[Figure]

**Fig. R10:** *Maps of 1979–2019 linear trends (Theil-Sen estimator; colour shading) in the seasonal RWP object frequency for (a) DJF, (b) MAM, (c) JJA, and (d) SON in ERA5. Red hatching indicates areas where the trend monotonicity is statistically significant at the 0.10 significance level.*

44. Caption of Fig. 7. Although written in the manuscript, consider saying in the caption of this figure that negative and positive values correspond to troughs and ridges for the purpose of clarity.

**Reply:** I agree; this reminder would help.
**Revision:** I have now included in the caption of Fig. 7 a reminder of what the trend sign denotes.

45. Lines 310-311. I suggest placing this sentence in the previous paragraph as it still discusses the Northern Hemisphere, and to start a new paragraph with the sentence "In contrast ...", addressing cp changes in the Southern Hemisphere.
**Reply:** Noted.
**Revision:** This is done as suggested (Lines 328–329; note that this sentence seems like a separate paragraph in the change-tracking version, but it is fine in the new clean manuscript version.).

46. Lines 311-313. Interesting, these results seem to show a coherent signal. Has this also been observed in previous studies? Do we understand why this change occurs? Please, see also general comment nr. 1.
**Reply:** Definite answers or even meaningful speculation in this "why" question are hardly possible. Clearly, the trends in the zonal wind field (Fig. S1) are a factor. To the best of my knowledge, no other study has dealt with past phase speed trends in the Southern Hemisphere upper-tropospheric circulation.

47. Lines 323-324. I cannot follow the sentence "In contrast to ... ... is defined". Please, rephrase.
**Reply:** I will.
**Revision:** This sentence has been clarified (Lines 344–345).

48. Lines 336-337. I have a bit of difficulty following the rather detailed scenarios sketched here. Isn't it the general idea of the analysis to assess the changes in the RWP amplitude-phase speed spectrum, that is, how slow to fast moving waves with small to large amplitudes change? Please, consider rephrasing to improve clarity of the text.
**Reply:** Lines 333–337 aim to explain the second reason (the first one is mentioned in lines 315–317) why assessing trends in $E$–$c_p$ bivariate PDFs can provide further information than what Figs. 6 and 8 alone suggest. An example is thus provided of an $E$–$c_p$ trend combination that may result from 2 different shifts in the bivariate PDF. I will try to make this clearer.
**Revision:** These two sentences have been removed. The new insights provided by this analysis is anyway presented in Lines 336–343.

49. Lines 341-342. There are several problems with the sentence "Although ... ... any E." For clarity, please, consider starting the sentence with "In C Asia ...", refer explicitly to Fig. 9f, and consider discussing the dipole of decreased waves with high cp of any E and increased waves with low cp of any E.
**Reply:** Noted. I will reformulate accordingly.
**Revision:** The sentence has changed accordingly (Lines 363–364).

50. Lines 339-347. These paragraphs are a bit hard to follow. Why are changes in DJF and MAM explained from the perspective of changes in E and cp, respectively, and not, for example, from both their changes? Is this because Figs 6 and 8 indicate largest changes in these seasons and regions for these specific measures respectively? If so, it may help the reader to state this. In addition, when speaking about the two N Pacific regions" (line 339) and "NE Pacific and N America" (line 343), it may be helpful for the reader to refer here to Figs 6a and 8b, respectively, to which these statements refer if I have at least understood the text correctly.
**Reply:** As mentioned in lines 337–338, not everything seen in Fig. 9 will be described in the text. The goal is to highlight the more evident/noticeable PDF shifts that perhaps provide further context in the robust trend patterns of Figs. 6 and 8. Even so, as the referee noticed, the text can get dense and hard to follow. Following the suggestion, I will try to better guide the reader in these paragraphs.
**Revision:** Several phrases in these two paragraphs have been reformulated. References to Figs. 6 and 8 have now been added and hopefully the story got clearer (Lines 360–370).

51. Line 348. To guide the reader, the sentence may start with "In JJA ...".
    **Reply:** Noted.
    **Revision:** The sentence has changed accordingly (Line 371).

52. Lines 16, 358, and 419; What is exactly meant with "no covariance is observed ...". Does the author perhaps mean to say that no "systematic changes" are observed in trends of the RWP amplitude and phase speed spectrum?
    **Reply:** An example may be helpful here. Both JJA in NE Pacific and MAM in Europe are characterized by a negative trend in $E$ (Fig. 9). However, in the first case there is a negative trend in $c_p$ and in the second case a positive trend in $c_p$. Moreover, JJA in Europe sees a negative $E$ trend, but no apparent trend in $c_p$. Overall, when considering all regions and seasons there is no systematic co-variability between the two properties in the past 4 decades. I will try to simplify these statements in the revised version.
    **Revision:** These examples have been added to the text in order to better illustrate this point (Lines 384–388).

53. Lines 360-361. I cannot follow the meaning of the sentence "Related to that ... ... in the Supplement of FW20)". Please, rephrase or consider deleting the sentence.
    **Reply:** This means to say that at weather time scales when e.g. $E$ is relatively low it doesn't mean that $c_p$ will be relatively high or low. There are several scenarios, which results in this triangular (or circular in some regions) shape in the black contours of Fig. 9 (see also reply to General Comment #7 of Referee #1). This is perhaps relevant to the fact that also at climate scales we see a variety of responses between the two properties in the 6 regions. I will rephrase to make this clearer.
    **Revision:** This sentence is actually not too relevant or necessary here, so it has been removed.

54. Line 388. I suppose the author means "not defined" or "undefined"?
    **Reply:** In Figs. 7, 8, and 10, grid points where a quarter or more of the seasons had no RWP object occurrence are masked (lines 273–276). In line 388, I make the distinction that for Fig. 11 the number of extremes in a season that had no RWP object occurrence will be 0. An argument could be made that this should be "NaN" (i.e., undefined), but I think 0 is more logical when counting events. In contrast, it would not be correct to assume that the phase speed is 0 when there is no RWP.

55. Lines 405-406. Consider writing "... seasonal mean E value in DJF, but not in JJA..."
    **Reply:** Noted.
    **Revision:** The sentence has changed accordingly (Line 437).

56. Lines 421-422. What is the implication of this finding?
    **Reply:** See my reply to General Comment #1.

57. Line 434. Probably, write "when" instead of "where"?
    **Reply:** Noted.
    **Revision:** This sentence is now changed (Lines 469–471).

**Bibliography**

Fragkoulidis, G. and Wirth, V.: Local rossby wave packet amplitude, phase speed, and group velocity: Seasonal variability and their role in temperature extremes, Journal of Climate, 33, 8767–8787, https://doi.org/10.1175/JCLI-D-19-0377.1, 2020.

Riboldi, J., Lott, F., D'Andrea, F., and Rivière, G.: On the Linkage Between Rossby Wave Phase Speed, Atmospheric Blocking, and Arctic Amplification, Geophysical Research Letters, 47, e2020GL087 796, https://doi.org/10.1029/2020GL087796, 2020.

Wolf, G. and Wirth, V.: Diagnosing the Horizontal Propagation of Rossby Wave Packets along the Midlatitude Waveguide, Monthly Weather Review, 145, 3247–3264, https://doi.org/10.1175/MWR-D-16-0355.1, 2017.

---

## Referee Report (RR1)

**Review of revised manuscript "wcd-2022-28"**

Title: Decadal variability and trends in extratropical Rossby wave packet amplitude, phase, and phase speed

Author: Georgios Fragkoulidis

**Summary**
As requested, I reviewed the revised manuscript along with the author responses. I found that the manuscript has been improved on several aspects, while the author also clarified raised comments and as to why to refrain from some proposed revisions. At the same time, I also feel that the improvements have been done partially, and that several of my comments need to be reconsidered along with additional changes in the manuscript. After implementing these changes, or clearly motivating why these changes are not desired, I recommend the manuscript to be accepted for publication. Below I outline these open points in a few general and minor comments.

**General comments**

**1. Motivation of the study, and the interpretation and implications of the results**
The revised manuscript improved by including a statement on the implications of the work in the abstract. Still, I feel that, after another careful read of the entire revised manuscript, that the main message of the article remains somewhat hidden. Here I do not suggest extending the material of the manuscript in other directions as perhaps unintentionally suggested in my earlier review. After re-reading the manuscript, I still wonder, what is the key message of this work that readers should take away? Perhaps, that decadal variability and trends of RWP characteristics vary substantially across regions and seasons? This may be important information in context of regional circulation changes in a warming climate and weather extremes (as indeed addressed as motivation in the introduction). I would like to suggest including the main message of the work in the abstract, for example, by replacing the phrase "; a manifestation of the pronounced … … in some areas and season" (lines 19-20) – which I find rather unclear – by such a key message. Also, I think the key message should be articulated in the "Summary and concluding remarks".

In addition to this general comment, I also think that one of the key points of section 3.1 – the positive (negative) trend in E over the N Pacific in DJF (JJA), and the narrowing distribution and reduced E over the N Atlantic in DJF and JJA, respectively, (based on Fig. 5) deserves to be mentioned in the conclusions. If the author agrees, perhaps a sentence on this subject can be added at line 415, between the sentence ending with "…. underestimate E." and starting with "Focusing on the 1979-2019 …".

**2. Section 3.3**
Okay, thanks for the clarification and mentioned text revisions.

**3. Section 3.4**
Thank you for the clarification on the analysis using two 20-year periods. About the (compound) extremes in other seasons, it is great these figures are added in the Supplement. However, in my opinion, the results for the other seasons should be briefly described in ±1

paragraph in section 3.4, and should also include a reason on why the author decided to elaborate on DJF, and why the other seasons were less interesting. Currently, as a reader I feel left behind with questions as to why DJF is chosen, and wonder how the analysis looks like for the other seasons. In my opinion, a scientific article shouldn't just add figures in a supplement without describing those in the text and leaving it up to the reader to interpret these figures.

**4. Decadal variability or trends? Section titles**

After re-reading the manuscript, I still felt sometimes somewhat confused whether the manuscript – as well as specific sections – address decadal variability or trends. In my opinion there is some inconsistency in the manuscript text:

- please, write "aspects of decadal variability and trends" in line 62 as the manuscript clearly addresses both;
- decadal variability and trends of the RWP in line 75 as subsection 3.1 addresses both;

Along the same lines, several section headers/titles do not adequately describe the context of the sections

- section 3.1; please, consider writing "Decadal variability and trends …" as trends are an important theme of this subsection (line 190);
- section 3.2; perhaps, consider "Spatial distribution of decadal …" (line 249);
- section 3.3 investigates "trends" and not the "variability" as it seems to me; please, consider writing "trends in joined Rossby wave packets amplitude and phase speed" (line 323)

**Minor comments**

Lines 4 and 204. To the author's reply on the comments using the phrase "may creep behind"; I understand perfectly what the author means. However, I do believe another phrasing is easily possible, for example, by simply saying "… to unveil past trends and interannual/decadal variability in the probability distribution of Rossby wave packet (RWP) amplitude … and phase speed (cp)." (Lines 3-4) and "… aim to highlight decadal variability against 'noise' from interannual variability" (Lines 204) or something along those lines. I do not mean to impose these specific suggestions but would like to encourage the author to consider rephrasing.

Lines 6-7. Please, consider simplifying the writing, for example, by saying "… where two historical reanalyses systematically underestimate E compared to three modern-era reanalyses."

Line 30. What would the sentence loose by removing the word "anyway"?

Caption of Fig. 7. Please, clarify in the caption whether the solid and dashed black contours depict positive or negative v.

Lines 286, 300, 304, etc. Please, consider rephrasing "frequency of occurrence of …" by "occurrence frequencies of …", here and elsewhere, which would read better.

Line 415. Please, consider writing "… but less in JJA where the historical reanalyses systematically …".

Line 418. Please, remove the phrase "The decadal variability of mean", and consider starting the sentence as is with "RWP properties …" since this paragraph - summarizing section 3.2 - discusses trends and not the variability.

Line 429. In my opinion, it seems more accurate to write "… are associated with varying shifts in the E-cp domain between seasons and regions" and to replace "a lack of" by "the absence of".

---

## Author Response (AR2)

**Decadal variability and trends in extratropical Rossby wave packet amplitude, phase, and phase speed (DOI: 10.5194/wcd-2022-28)**

**Reply to referee comments on the revised manuscript**

Georgios Fragkoulidis (JGU Mainz)

Once more I would like to thank the two referees for kindly reviewing the revised version of the manuscript. I appreciate their time and I am glad that they found this version improved. Below are point-by-point replies (in black) to their comments (in blue) as well as a few additional changes I deemed necessary. Where applicable, line numbers and figure labels in my replies correspond to the marked-up version of the revised manuscript.

**Referee #1 comments**

I appreciate the detailed answers to the raised points, which will remain public for the potentially interested readers, and I am overall very satisfied with the revised version of the paper.

I just have a few, minor points for further consideration of the author before publication:

1. Reply to major point 4: A sentence could be added confirming that results of the trend analysis (Fig. 8) do not change if the angular velocity is used instead of the linear velocity.
   **Reply:** This is now mentioned (Lines 331–333)

2. Reply to major point 5 and 7: From the provided answer, it seems that atmospheric blocking would follow indeed the expected high E/low cp relationship. However, I still find puzzling that such a relationship does not appear in the analysis leading to Fig. 9 (and in FW20): is wave breaking leading to a decay in E but not necessarily of cp? I realize that this curiosity likely deserves a separate study, unless the author has additional comments on the above point. However, parts of the reply or references to relevant results by FW20 could be added to further explain this "surprising" result. A similar consideration holds for the thorough reply to major point 7, maybe a part of it can be added to the main manuscript to further contextualize the lack of covariance.
   **Reply:** Cases of blocking are only a small subset of all days considered in this analysis and the fact that $E$ and $c_p$ are not generally anticorrelated (as indicated by the shape of the climatological-mean $E$–$c_p$ spectra) is not too surprising. For example, there can be low-amplitude RWP objects (these are still features with an amplitude of above $15ms^{-1}$) with low, average or high phase speed. The reason is that these two RWP properties do not just depend on each other, but on other factors as well. The absence of covariance in $E$ and $c_p$ trends of various regions and seasons is arguably a manifestation of that. I added comments along these lines in the main text and hopefully the issue is now clearer (Lines 377–381).

3. Reply to major point 6: I do not immediately see where the issue has been discussed as the line range is unclear (the paper ends at line 470, maybe a typo?).
   **Reply:** This is not a typo. Line numbers in my replies refer to the marked-up version of the

manuscript, such that the old and new text is seen at once.

4. Line 19: verb is missing, should it read as "This is a manifestation"?
   **Reply:** I added a verb as suggested (Line 20).

5. Line 180: shouldn't it read "seasonal means"?
   **Reply:** Both plural and singular forms are fine here. As it stands, it denotes the standard deviation of a variable (the field's seasonal mean in this case). If the variable is thought of as a set of data points, then "seasonal means" should be used.

6. Line 219: "might not be well constrained", as it is only a (reasonable and interesting) speculation?
   **Reply:** The speculative nature of this statement was implied by the verb "suggests" in the beginning of the sentence. Nevertheless, I followed the referee's suggestion to make this clearer (Line 223).

**Referee #2 comments**

**Summary**

As requested, I reviewed the revised manuscript along with the author responses. I found that the manuscript has been improved on several aspects, while the author also clarified raised comments and as to why to refrain from some proposed revisions. At the same time, I also feel that the improvements have been done partially, and that several of my comments need to be reconsidered along with additional changes in the manuscript. After implementing these changes, or clearly motivating why these changes are not desired, I recommend the manuscript to be accepted for publication. Below I outline these open points in a few general and minor comments.

**General comments**

1. **Motivation of the study, and the interpretation and implications of the results**
   The revised manuscript improved by including a statement on the implications of the work in the abstract. Still, I feel that, after another careful read of the entire revised manuscript, that the main message of the article remains somewhat hidden. Here I do not suggest extending the material of the manuscript in other directions as perhaps unintentionally suggested in my earlier review. After re-reading the manuscript, I still wonder, what is the key message of this work that readers should take away? Perhaps, that decadal variability and trends of RWP characteristics vary substantially across regions and seasons? This may be important information in context of regional circulation changes in a warming climate and weather extremes (as indeed addressed as motivation in the introduction). I would like to suggest including the main message of the work in the abstract, for example, by replacing the phrase "; a manifestation of the pronounced ... ... in some areas and season" (lines 19-20) — which I find rather unclear — by such a key message. Also, I think the key message should be articulated in the "Summary and concluding remarks".

   In addition to this general comment, I also think that one of the key points of section 3.1 — the positive (negative) trend in E over the N Pacific in DJF (JJA), and the narrowing distribution and reduced E over the N Atlantic in DJF and JJA, respectively, (based on Fig. 5) deserves to be mentioned in the conclusions. If the author agrees, perhaps a sentence on this subject can be added at line 415, between the sentence ending with ".... underestimate E." and starting with "Focusing on the 1979–2019 ..."
   **Reply:** Apart from listing the key outcomes of the individual analyses it is indeed worth concluding with a couple of sentences that extract a single take-home message. The referee is right about the key message in this case. The final sentence of the abstract has now changed to such a

statement (Lines 21–24), rather than the generic comment of the prior version. The key message of the study is now also articulated at the end of the summary in section 4 (Lines 463–465). In addition, the outcomes of Fig. 5 are now included in the conclusions as per the suggestion of the referee (Lines 442–444).

2. **Section 3.3**
Okay, thanks for the clarification and mentioned text revisions.

3. **Section 3.4**
Thank you for the clarification on the analysis using two 20-year periods. About the (compound) extremes in other seasons, it is great these figures are added in the Supplement. However, in my opinion, the results for the other seasons should be briefly described in $\pm 1$ paragraph in section 3.4, and should also include a reason on why the author decided to elaborate on DJF, and why the other seasons were less interesting. Currently, as a reader I feel left behind with questions as to why DJF is chosen, and wonder how the analysis looks like for the other seasons. In my opinion, a scientific article shouldn't just add figures in a supplement without describing those in the text and leaving it up to the reader to interpret these figures.
**Reply:** Section 3.4 has been updated to address these issues. The analysis is restricted to Northern Hemisphere DJF, since one of the two goals of this subsection (see Lines 385–387) is just to emphasize that temporal variations in the aforementioned trends may exist. The pronounced interannual-to-decadal variability over N Pacific in DJF provides a good archetype that serves this purpose without tiring the reader too much. Nevertheless, it is worth adding the corresponding analysis of the other seasons in the Supplement for reference and, indeed, worth commenting on them in the main text. I have now added a paragraph that briefly discusses their main points (Lines 421–431).

4. **Decadal variability or trends? Section titles**
After re-reading the manuscript, I still felt sometimes somewhat confused whether the manuscript — as well as specific sections — address decadal variability or trends. In my opinion there is some inconsistency in the manuscript text:
• please, write "aspects of decadal variability and trends" in line 62 as the manuscript clearly addresses both;
• decadal variability and trends of the RWP in line 75 as subsection 3.1 addresses both;

Along the same lines, several section headers/titles do not adequately describe the context of the sections
• section 3.1; please, consider writing "Decadal variability and trends ..." as trends are an important theme of this subsection (line 190);
• section 3.2; perhaps, consider "Spatial distribution of decadal ..." (line 249);
• section 3.3 investigates "trends" and not the "variability" as it seems to me; please, consider writing "trends in joined Rossby wave packets amplitude and phase speed" (line 323)
**Reply:** I have followed the specific referee's suggestions in order to minimize this inconsistency and better reflect the sections' contents. A small deviation is that Section 3.3 is now titled: "Trends in the Rossby wave packet amplitude and phase speed joint distribution".

**Minor comments**

1. Lines 4 and 204. To the author's reply on the comments using the phrase "may creep behind"; I understand perfectly what the author means. However, I do believe another phrasing is easily possible, for example, by simply saying "... to unveil past trends and interannual/decadal variability in the probability distribution of Rossby wave packet (RWP) amplitude and phase speed

(cp)." (Lines 3–4) and "... aim to highlight decadal variability against "noise" from interannual variability" (Lines 204) or something along those lines. I do not mean to impose these specific suggestions but would like to encourage the author to consider rephrasing.

**Reply:** The two instances of "creep" have now been replaced (Lines 4 and 207)

2. Lines 6-7. Please, consider simplifying the writing, for example, by saying "... where two historical reanalyses systematically underestimate E compared to three modern-era reanalyses."

**Reply:** Changed as suggested (Lines 6–8)

3. Line 30. What would the sentence loose by removing the word "anyway"?

**Reply:** The word "anyway" here emphasizes the fact that any forcing on the circulation due to global warming (as well as any effect the circulation has on the temperature field) may lead to variability and trends on top of the ones already generated owing to internal natural variability.

4. Caption of Fig. 7. Please, clarify in the caption whether the solid and dashed black contours depict positive or negative v.

**Reply:** Since contour labels are perhaps not clear, I have added this information in the captions of Figs. 7 and 8.

5. Lines 286, 300, 304, etc. Please, consider rephrasing "frequency of occurrence of ..." by "occurrence frequencies of ...", here and elsewhere, which would read better.

**Reply:** These instances have changed as suggested.

6. Line 415. Please, consider writing "... but less in JJA where the historical reanalyses systematically ...".

**Reply:** This has now changed (Line 440).

7. Line 418. Please, remove the phrase "The decadal variability of mean", and consider starting the sentence as is with "RWP properties ..." since this paragraph — summarizing section 3.2 — discusses trends and not the variability.

**Reply:** This sentence has been reformulated to account for this (Lines 445–447).

8. Line 429. In my opinion, it seems more accurate to write "... are associated with varying shifts in the E–cp domain between seasons and regions" and to replace "a lack of" by "the absence of".

**Reply:** This is true. I have changed that as suggested (Lines 456–458).

**Additional changes**

1. A sentence is added to explain the meaning of the seasonal-mean phase index value with an example (Lines 282–284).

2. I changed "multi-decadal" to "decadal" in order to have a consistent terminology for the trend analyses in the text (Line 210).

3. I changed "most" to "much" (Line 394).

4. I moved the reference to Figs. S6–S8 from Line 423 to 410.

5. Line 494 has been simplified.

---

## Author Response (AR3)

**Decadal variability and trends in extratropical Rossby wave packet amplitude, phase, and phase speed (DOI: 10.5194/wcd-2022-28)**

**Reply to the Co-Editor decision**

Georgios Fragkoulidis (JGU Mainz)

I would like to thank the co-editor for accepting my manuscript for publication, subject to technical corrections. I appreciate the time spent to coordinate the review process toward this. For the final submission, I have reformulated some bits of the text in order to account for the two corrections requested by the Co-Editor. Firstly, there is a slight rearrangement in the sentence of Lines 453–455 that makes the syntax simpler. In addition, while correcting the caption of Fig. 9, I realised that the term "$E$–$c_p$ distribution" that contains an en dash between $E$ and $c_p$ may lead to confusion (e.g., that it denotes the "$E$ minus $c_p$ distribution", which is not the case). I therefore introduced the symbol $f_{E,c_p}$ in Line 334 and used it in Lines 340, 347, 361, and 371 in order to avoid this potential confusion in the respective sentences. Related to that, the term "$E$–$c_p$ distribution" in Lines 17 and 398 has been replaced by "$E$–$c_p$ domain" which is harder to misunderstand, while Line 446 is also slightly simplified.